# Single-cell new RNA sequencing reveals principles of transcription at the resolution of individual bursts

**Daniel Ramsköld** [1,2], **Gert-Jan Hendriks**[1,2], **Anton J. M. Larsson** [1,2], **Juliane V. Mayr** [1], **Christoph Ziegenhain** [1], **Michael Hagemann-Jensen** [1], **Leonard Hartmanis** [1] & **Rickard Sandberg** [1] ✉

Analyses of transcriptional bursting from single-cell RNA-sequencing data have revealed patterns of variation and regulation in the kinetic parameters that could be inferred. Here we profiled newly transcribed (4-thiouridine-labelled) RNA across 10,000 individual primary mouse fibroblasts to more broadly infer bursting kinetics and coordination. We demonstrate that inference from new RNA profiles could separate the kinetic parameters that together specify the burst size, and that the synthesis rate (and not the transcriptional off rate) controls the burst size. Importantly, transcriptome-wide inference of transcriptional on and off rates provided conclusive evidence that RNA polymerase II transcribes genes in bursts. Recent reports identified examples of transcriptional co-bursting, yet no global analyses have been performed. The deep new RNA profiles we generated with allelic resolution demonstrated that co-bursting rarely appears more frequently than expected by chance, except for certain gene pairs, notably paralogues located in close genomic proximity. Altogether, new RNA single-cell profiling critically improves the inference of transcriptional bursting and provides strong evidence for independent transcriptional bursting of mammalian genes.

It has been nearly 50 years since the transcription of nascent RNA was described as a bursting process, where periods of transcriptional activity were interspersed with periods of inactivity[1]. Direct evidence of stochastic transcription and bursting dynamics have come from real-time imaging experiments where nascent RNAs are monitored over time[2–4] and indirectly from time-lapse microscopy on the resulting protein levels[5]. Complementing evidence has emerged from analyses of steady-state RNA counts across single cells, either using single-molecule RNA fluorescence in situ hybridization (smRNA-FISH)[6–8] or single-cell RNA sequencing (scRNA-seq)[9] together with modelling to infer kinetic parameters that best describe the observed RNA count distributions. Whereas both strategies can summarize average bursting features, for example, the burst frequency or burst size (that is, RNAs transcribed per burst), real-time methods can investigate the variation in bursting over time. The strength of scRNA-seq-based bursting inferences lies in the ability to infer allele-level kinetics across thousands of endogenous genes in parallel, whereas more targeted smRNA-FISH approaches have higher sensitivity, which is important for accurate burst size estimations.

Although transcriptional bursting has been extensively studied[10,11], there are several important open questions regarding bursting kinetics. A central question is whether all genes are transcribed in bursts or whether subsets of genes also show constitutive expression. Most evidence argues for general transcriptional bursting, although

[1]Department of Cell and Molecular Biology, Karolinska Institute, Stockholm, Sweden. [2]These authors contributed equally: Daniel Ramsköld, Gert-Jan Hendriks, Anton J. M. Larsson. ✉e-mail: Rickard.Sandberg@ki.se

**Fig. 1 | High-quality profiling of new RNAs in single cells with NASC-seq2.**
**a**, Plot showing the number of genes detected per K562 cell as a function of reads sequenced, for K562 cells processed with NASC-seq2 (613 individual cells) and NASC-seq[20] (138 individual cells), respectively. The mean number of genes per method and sequencing depth is shown, together with error bars (1.96× s.e.m.). **b**, Illustration of large-scale NASC-seq2 experiment on F1 primary fibroblasts. Four technical replicates of primary fibroblast cultures were independently exposed to 4sU and collected for FACS and NASC-seq2 library construction. For transcriptional dynamics analyses, cells from all replicates were pooled. **c**, Uniform Manifold Approximation and Projection (UMAP) of primary fibroblasts, overlayed with contour plots, showing that assayed primary fibroblast cells did not show apparent patterns of heterogeneity. **d**, Boxplots showing the obtained signal-to-noise level (Pc/Pe) in fibroblasts with ($n$ = 8,912) and without ($n$ = 783) 4sU (2 h). The boxplots show the median and boundaries (first and third quartile), and the whiskers denote 1.5 times the interquartile range of the box. **e**, Density plot for the obtained power to call RNA molecules as new ($y$ axis) against the reconstructed RNA molecule length ($x$ axis). **f**, Contour plots showing the fraction of new RNA molecules per cell ($x$ axis) against total detected RNA molecules per cell ($y$ axis) for fibroblasts with and without 4sU. **g**, Scatter plot of burst frequency estimates ($x$ axis) for mouse primary fibroblasts previously inferred from total RNA counts[9] against the fraction of cells with new RNA ($y$ axis) detected after 2-h 4sU exposure. Source numerical data are available in Source data.

predominantly constitutive expression was observed in studies in bacteria[12] and human cells[13], and for specific genes in yeast[3,14]. While transcriptome-wide inference of bursting parameters from steady-state single-cell RNA counts is effective[9], the synthesis and closing rates (that together make up the burst size) could not be individually determined, and absolute burst frequency estimates required separately measured degradation rates. Therefore, information on how long bursts lasts is mostly derived from sporadic observations[11].

An equally fundamental question for transcription is whether the bursting of each gene is independent or whether closely related genes (by genomic or spatial distance) are prone to co-bursting. Several lines of evidence have indirectly implied coordinated co-bursting of multiple genes, such as reports of transcriptional factories[15,16] and transcriptional condensates[17]. A recent study found spatial coupling in nascent RNA when the gene loci are in close spatial proximity[18]. Intriguingly, studies in the fruit fly have demonstrated coordinated bursting of transgenes[4] and for pairs of paralogues[19]. However, allele-level analyses of co-bursting can fully control for spurious correlations from cellular heterogeneity (for example, cell cycle and activation states) or technical variability in measurements that otherwise could lead to false positive correlations.

In this study, we investigated transcriptional bursting kinetics transcriptome-wide in primary fibroblasts through temporally resolved scRNA-seq. Analysis of the newly transcribed RNA greatly improved the inference of kinetic parameters. Interestingly, the varying burst sizes (RNA molecules per burst) observed across genes were found to correlate with inferred synthesis rate, whereas the burst durations showed little variation across genes. Investigating the allele-level new RNA profiles across the single cells, we demonstrate an overall lack of co-bursting of nearby genes except for a few gene pairs with modest increase in co-bursting.

## Results

### Improved new RNA profiling in single mouse cells

An attractive strategy for analyses of transcriptional dynamics and bursting kinetics is to count only the RNA molecules transcribed within a defined time period, demonstrated by recent 4-thiouridine (4sU)-based single-cell new RNA profiling methods[20,21]. In these methods, cells are exposed to the uridine analogue 4sU for a short period of time, leading to 4sU being incorporated into the transcribed RNA. During library construction, the alkylation of 4sU and subsequent reverse transcription (RT) results in base conversions in the complementary DNA at the positions of the 4sU incorporation[20,21]. The presence of 4sU-induced T-to-C conversions against the reference genome enables the computational separation of new and pre-existing RNA molecules. Previous 4sU-based single-cell methods[20,21], however, suffered from low sensitivity and cellular throughput. Here, we developed NASC-seq2, a miniaturized version of NASC-seq[20] with higher sensitivity and cellular throughput, that also includes unique molecular identifier (UMI). To compare the performance of NASC-seq2 with

the original NASC-seq[20], we applied NASC-seq2 to individual K562 cells ($n = 613$) that were exposed to 4sU for 2 h. NASC-seq2 showed drastically improved sensitivity and detected on average 2,000 more genes per cell (at matched 100,000 total RNA reads) compared with NASC-seq (Fig. 1a). The improvement mainly stems from starting with nanolitre lysis volume (following Smart-seq3xpress[22]), which enabled the dimethyl sulfoxide (DMSO)-based alkylation step to be carried out in a low volume and subsequently diluted out (instead of using bead purification before cDNA amplification). Since the ability to separate new and old RNAs depends on the length of sequenced reads (Extended Data Fig. 1a), we used longer short-read sequencing strategies (PE200). Analyses of the observed base conversions, through a mixture model that infers the probability of 4sU-induced base conversions (Pc) and conversions due to errors introduced during library preparation or sequencing (Pe), demonstrated a high signal-to-noise (Pc/Pe) ratio of ~45 (Extended Data Fig. 1b–d). The average power in assigning new RNA molecules was above 90% (Extended Data Fig. 1e,f), and we found that approximately 20% of the detected RNA molecules in K562 cells were transcribed within the 2-h period (Extended Data Fig. 1g,h).

## Analysis of transcriptional dynamics in mouse primary fibroblasts

We next sought to create a comprehensive transcriptome dynamics data set, by applying NASC-seq2 to profile new RNA (2-h 4sU labelling) on 10,000 individual primary mouse fibroblasts (Fig. 1b). The primary fibroblasts came from a female F1 mouse (C57Bl/6 × CAST) so that transcribed genetic polymorphisms could be used to also assign RNA to the alleles[23]. Single-cell libraries from 8,912 4sU-exposed cells passed quality control, and analysed cells were homogeneous and with no substructures in lower-dimensional projections (Fig. 1c). The median signal-to-noise ratio was 20, showing strong separation between 4sU-exposed cells ($n = 8,912$) and control non-exposed fibroblast cells ($n = 783$) (Fig. 1d and Extended Data Fig. 2). The average power to call new RNA molecules was 70% and dependent on sequenced and reconstructed length (Fig. 1e and Methods). We detected approximately 100,000 RNA molecules per cell, out of which 12.5% were assigned as new (Fig. 1f). Comparing the fraction of cells with detected new RNA to previously reported burst frequencies for similar fibroblasts[9] revealed a strong correlation (Spearman $r = 0.9$) (Fig. 1g), indicating that the observed new RNA counts were in general agreement with transcriptome-wide transcriptional burst kinetic data inferred from steady-state scRNA-seq[9].

The two-state telegraph model of transcription[24] (Fig. 2a) is often used for steady-state estimation of kinetics, where four rate parameters dictate the transcriptional dynamics. Each loci transition from transcriptional off or on state (based on the $k_{on}$ and $k_{off}$ rates), where the rate of RNA transcription in the on state is controlled by the synthesis rate ($k_{syn}$) and subject to RNA degradation ($k_d$). To extend the model to the transient (pulse-labelling) state, the probability mass function was derived that describe the new RNA counts as a function of bursting kinetic parameters and 4sU-labelling time (Methods and Supplementary Note). Having measured both new and pre-existing RNA per cell enabled us to also derive degradation rates. Using the mass function and degradation rates, three gene-level transcriptional bursting parameters were inferred from new RNA counts using maximum likelihood estimation, with parameters initialized from three count summary statistics (Methods). These included the fraction of cells with new RNA (Fig. 2b) and the coefficient of variation in new RNA counts (Fig. 2c), which we found informative primarily for $k_{on}$ and $k_{syn}$, respectively. The distribution of all four inferred rate parameters was visualized on absolute time scales (Fig. 2d) with indicated number of genes robustly inferred per distribution and error bars with geometric standard deviation showing accuracy. Typical half-lives of transcripts were between an hour and a day, and the frequency of bursts varied from one per day to one per hour, whereas a burst lasted only around

a minute during which RNA can be transcribed at rates of 3–200 molecules per hour (Fig. 2d).

However, a general problem with the joint inference of kinetic parameters is the risk of spurious correlations between parameters. In fact, the synthesis rate ($k_{syn}$) and off rate ($k_{off}$) were found to correlate when inferred from the 8,912 fibroblast cells (Fig. 2e), probably reflecting technical noise during inference affecting both parameters. To this end, we separated the cells into two halves and performed independent inference on each half. Importantly, the correlation between the synthesis rate ($k_{syn}$) and off rate ($k_{off}$) was strongly reduced (Fig. 2f). The inference of individual parameters from the two cell halves was reproducible (Fig. 2g–j), with the highest number of cells required for the inferences of synthesis ($k_{syn}$) and off ($k_{off}$) rates (Fig. 2k). Moreover, simulations were used to validate the correct central estimates and unimodal distribution (Extended Data Fig. 3a–c). Reassuringly, burst frequency and size estimates inferred from steady-state scRNA-seq data were highly concordant with those inferred from new RNA profiles (Extended Data Fig. 3d,e). Thus, inference from steady-state scRNA-seq data fails to accurately infer transcriptional off ($k_{off}$) and synthesis rate ($k_{syn}$) (Extended Data Fig. 3), but those parameters could be inferred from new RNA counts (Fig. 2).

Having determined the robustness of the inference based on the analyses of cell halves, we next explored patterns of bursting kinetics from the transcriptome-wide data. Interestingly, the transcriptional off ($k_{off}$) rates were 100-fold higher than transcriptional on ($k_{on}$) rates, demonstrating that all inferred genes were expressed in bursts. Focusing on the smaller subset of 1,216 genes for which all four parameters were reproducibly inferred in both cell halves, we found that the rate constants ($k_{on}$ and $k_{off}$) were uncorrelated even though they specify a mutually reversible process (Fig. 3a). Interestingly, we found that only the synthesis rate ($k_{syn}$) was correlated with inferred burst size, whereas $k_{off}$ was not, indicating that the burst duration is more invariant while the rate of synthesis specifies the amounts of RNA produced per burst (Fig. 3b,c). We validated that the synthesis rate ($k_{syn}$) controls the burst size, through similar kinetic inference in the K562 cells (Extended Data Fig. 4) albeit with lower correlation and accuracy due to fewer sequenced cells. Analysis of core promoter elements revealed significant interactions with burst size (and not frequency), as previously reported[9], and additionally the interaction was found for $k_{syn}$ since it determines the burst size (Extended Data Fig. 5). A systematic comparison of measured and derived parameters across 7,234 genes demonstrated the impact of burst frequency on the overall expression and that the correlation between burst size and synthesis rate holds transcriptome-wide (Fig. 3d). In line with the small burst sizes detected, which in part can be a technical underestimate, we find a moderate correlation between burst size and the fraction of cell with new RNA, possibly indicating that unproductive on states may occur. Highly expressed genes are more variable in terms of burst size, whereas medium and lowly expressed genes differ mostly in burst frequency (Fig. 4a–c), as previously reported[5]. Correlating observed and inferred parameters in highly expressed genes revealed a negative correlation between degradation rate and burst frequency (Fig. 4d).

## Transcriptome-wide analysis of co-bursting at allelic resolution

Studies have reported gene pairs showing co-bursting[4,18,19], that is, that simultaneous RNA transcription occurred more often than expected by chance (Fig. 5a). However, no previous study has investigated allele-level measurements, which constitute an important control, where nearby genes on the same parental chromosome (allele) can be compared with comparisons between the same gene pair on the opposite chromosome (other allele). Co-bursting should manifest as genes that burst more often from the same allele across cells than expected if all genes were transcribed independently (Fig. 5b–d). Having performed the experiments in F1 mice, we assigned all new RNA molecules (and reads) to their

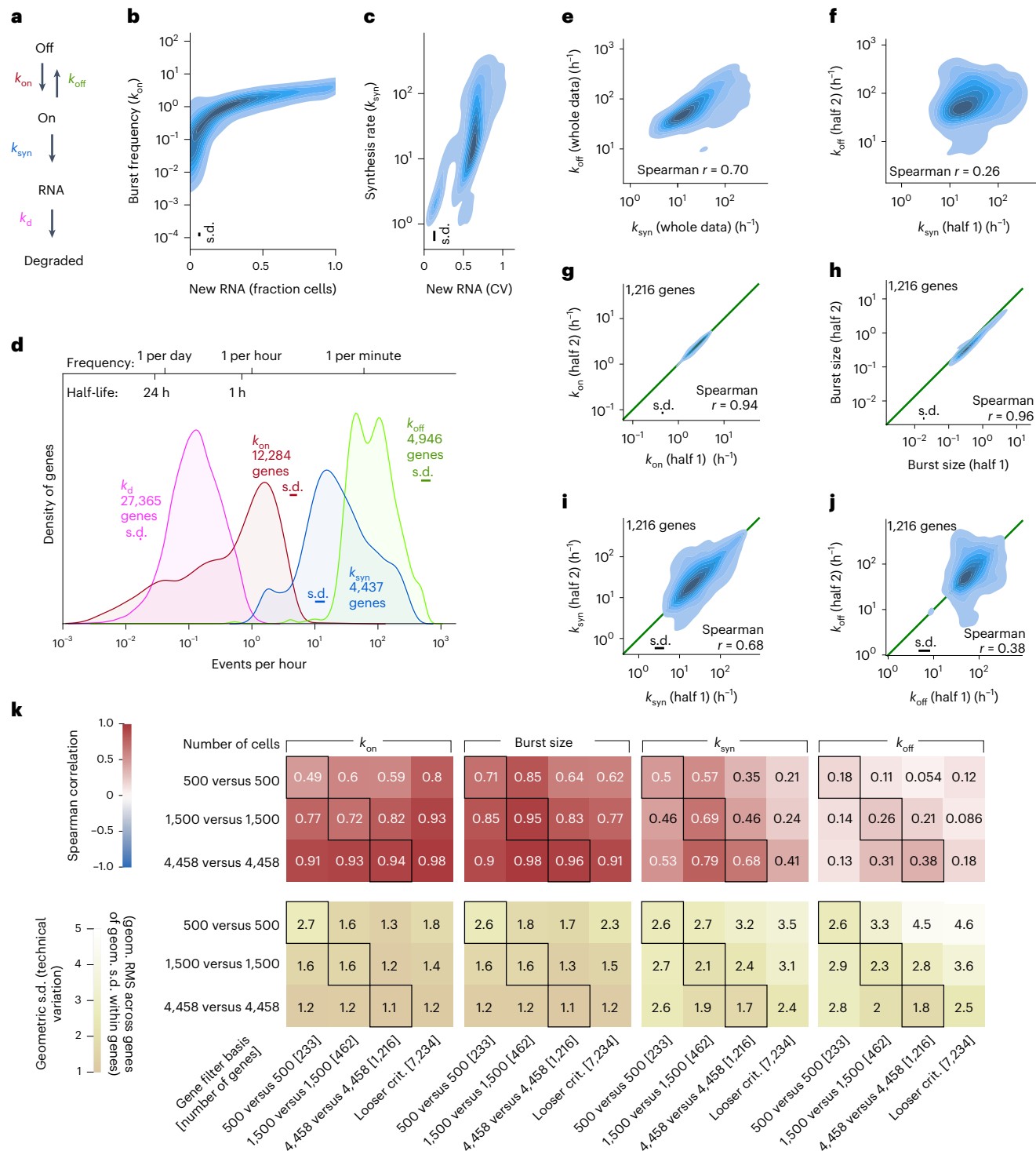

**Fig. 2 | Transcriptional burst kinetics inference from single-cell new RNA counts. a**, Illustration of the two-state telegraph model of transcription. **b**, Contour plot of inferred burst frequencies ($k_{on}$) against the fraction of cells with detected new RNA (12,284 genes included with robust $k_{on}$ inference). **c**, Contour plot of synthesis rate ($k_{syn}$) against the coefficient of variation (CV) of new RNA counts over cells (4,437 genes with robust $k_{syn}$ inference). **d**, Density plots for inferred bursting parameters in primary fibroblasts, with the number of genes for which the respective parameter could be robustly assigned as shown in the figure. Technical losses could cause a constant $k_{syn}$ underestimation bias. Top bar: indicative waiting times. **e**, Contour plot of synthesis ($k_{syn}$) rate against transcriptional off ($k_{off}$) rate inferred from all primary fibroblasts. **f**, Contour plot of synthesis ($k_{syn}$) rates inferred on cell subset half 1 against transcriptional off ($k_{off}$) rates inferred from cell subset half 2. **g**, Contour plot of

burst frequency ($k_{on}$) rates inferred on cell subset half 1 against burst frequency ($k_{on}$) rates inferred from cell subset half 2. **h**, Contour plot of burst size ($k_{syn}/k_{off}$) inferred on cell subset half 1 against burst size ($k_{syn}/k_{off}$) rates inferred from cell subset half 2. **i**, Contour plot of synthesis ($k_{syn}$) rates inferred on cell subset half 1 against synthesis ($k_{syn}$) rates inferred from cell subset half 2. **j**, Contour plot of transcriptional off ($k_{off}$) rates inferred on cell subset half 1 against transcriptional off ($k_{off}$) rates inferred from cell subset half 2. Plots in **d**–**g** are based on 1,216 genes that had robust inference on all four parameters in each cell subset. **k**, Correlation matrices, summarizing Spearman correlations obtained when comparing inferences from the two cell subsets, but subsampling the numbers of cells per subset and ordering the genes according to their mean expression. Geometric standard deviation (s.d., technical variation) is shown as error bars. Source numerical data are available in Source data.

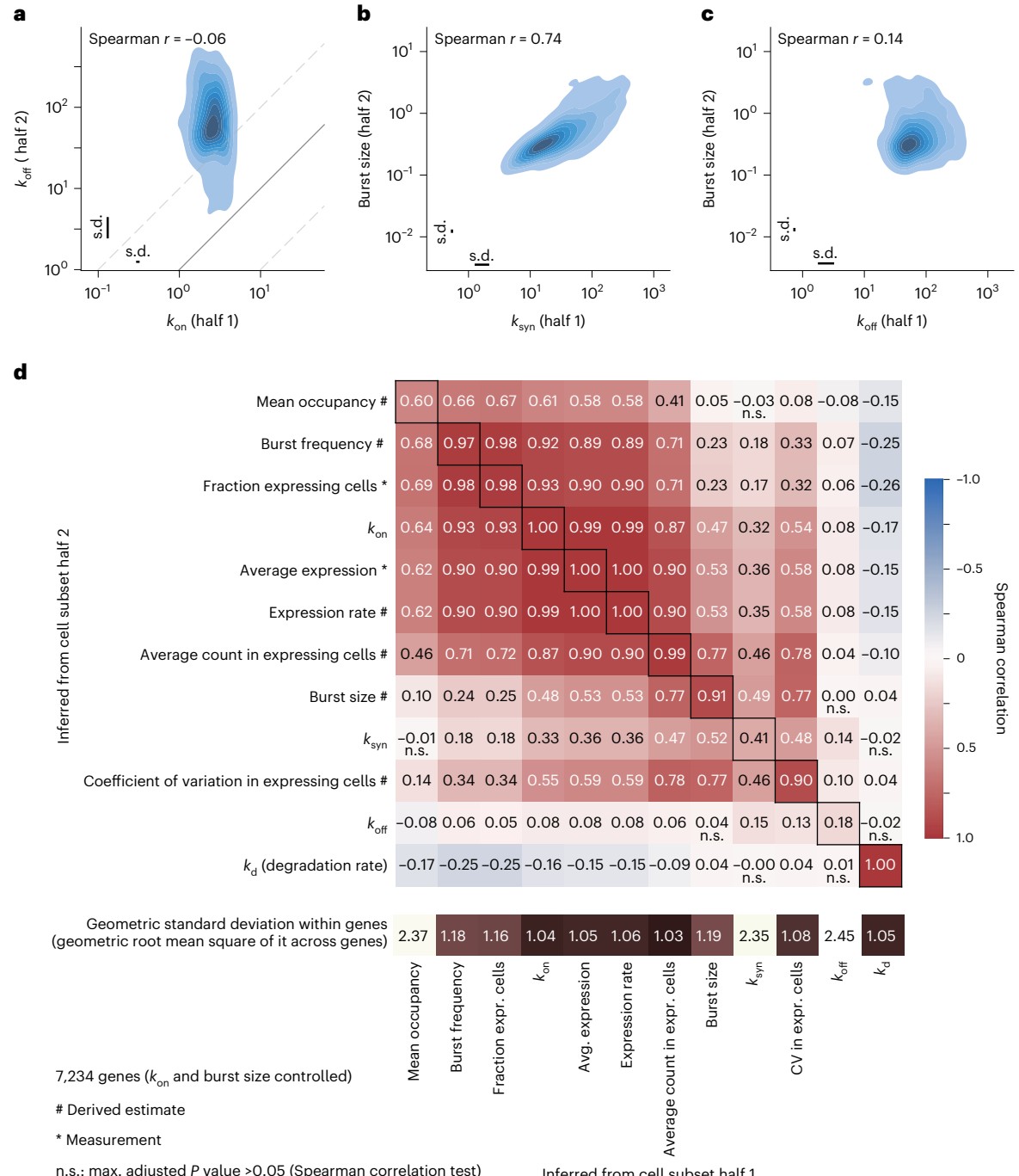

**Fig. 3 | Burst size is controlled by synthesis rate. a**, Contour plot of transcriptional on ($k_{on}$) and off ($k_{off}$) rates, inferred separately on two non-overlapping halves of the cells (1,216 genes with robust inference of $k_{on}$, $k_{off}$, $k_{syn}$ and burst size). **b**, Contour plot of burst size (inferred from cell subset half 2) against the synthesis rate, $k_{syn}$ (inferred from cell subset half 1) (1,216 genes with robust inference of $k_{on}$, $k_{off}$, $k_{syn}$ and burst size). **c**, Contour plot of burst size (inferred from cell subset half 2) against the off rate, $k_{off}$ (inferred from cell

subset half 1) (1,216 genes with robust inference of $k_{on}$, $k_{off}$, $k_{syn}$ and burst size). **d**, Spearman correlation matrix from parallel inference of two cell halves (each with 4,458 cells) based on the 7,234 genes with robust $k_{on}$ and burst size inference. Measurements are indicated with asterisks in contrast to derived estimates (#). $P$ values from the Spearman correlation tests were Benjamini–Hochberg adjusted. In **a**–**c**, geometric standard deviation (s.d., technical variation) is shown as error bars. Source numerical data are available in Source data. expr.,expressing.

allelic origin (Methods). As previously shown[23,25], the allelic estimates accurately capture X-chromosome inactivation and imprinting features (Extended Data Fig. 6). We computed allelic new RNA counts for each gene and cell and performed pair-wise comparisons of genes as a function of their genomic distance (on the linear chromosome).

Through Gillespie bursting simulation using the inferred gene-level kinetics, we simulated new RNA counts and monitored the

fraction of cells with new RNA observations and how often new RNA observations were derived from a single burst event (Methods). We computed the average detection of new RNA events (across expressed genes) across cells labelled for different time periods, and the fraction of those new RNA observations that were derived from a single burst event (Fig. 5e). The analysis demonstrated that 4sU labelling for 1–2 h led to the highest frequency of single-burst new RNA observations

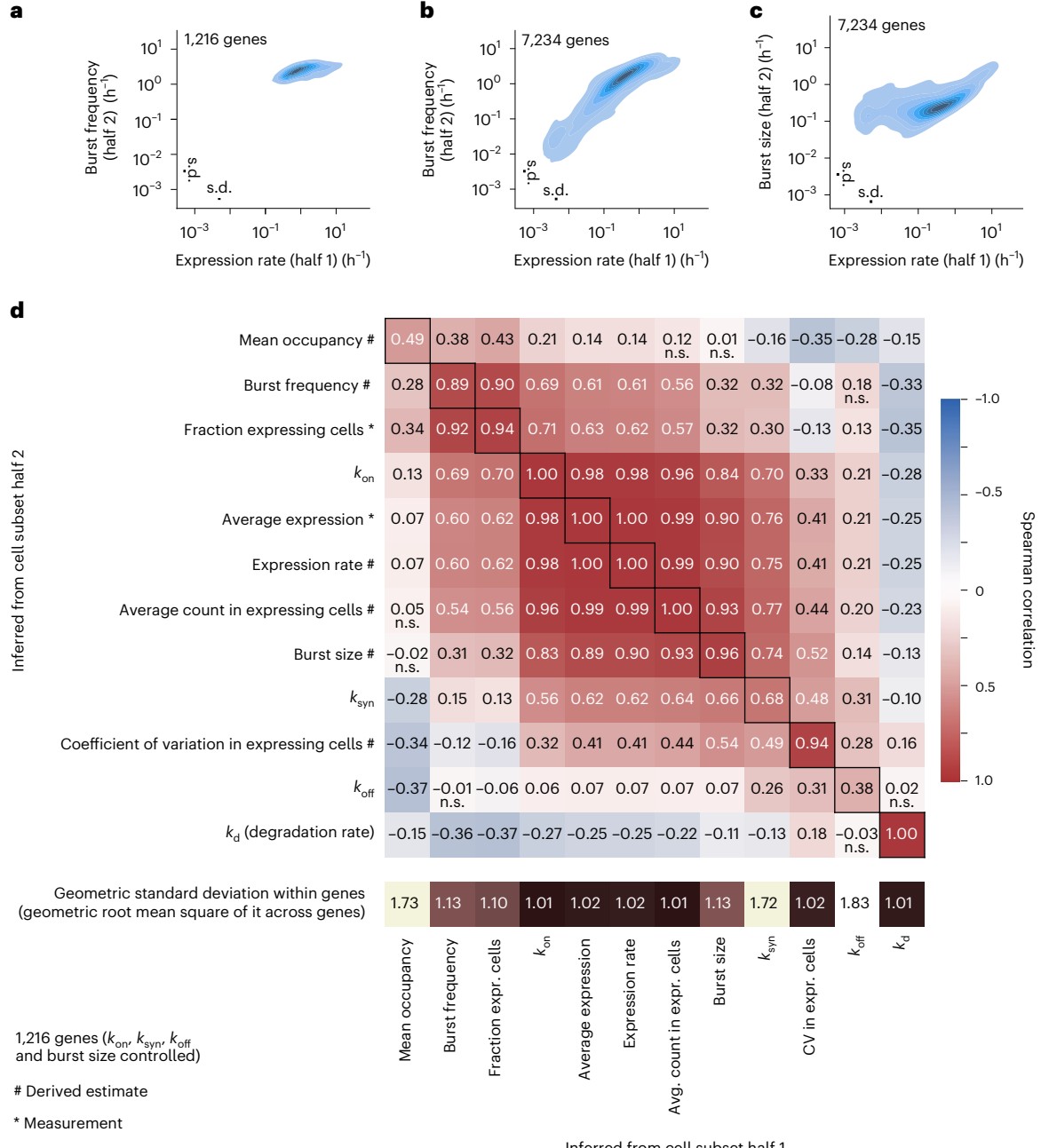

**Fig. 4 | Transcriptional burst kinetic analysis of highly expressed genes.**
**a,b**, Contour plot of expression rate (inferred from cell subset half 1) against burst frequency (inferred from cell subset half 2), for the 1,216 most highly expressed genes (**a**) or for 7,234 genes (**b**). **c**, Contour plot of expression rate (inferred from cell subset half 1) against burst size (inferred from cell subset half 2) for 7,234 genes. **d**, Correlation matrix (as in Fig. 3d) for the 1,216 most highly expressed genes (4,458 cells per cell half). Apart from the four estimated parameters in

the telegraph model, the heat map includes measurements (*) and parameter-derived estimates (#), where mean occupancy (fraction of time in on state) = $k_{on}/(k_{on} + k_{off})$, burst size (RNAs per on state) = $k_{syn}/k_{off}$, burst frequency (on states per time) = $1/(1/k_{on} + 1/k_{off})$, expression rate (RNAs per time) = $k_{syn} \times k_{on}/(k_{on} + k_{off})$. Geometric standard deviation indicates technical variation. n.s., $P > 0.05$. Source numerical data are available in Source data.

(Fig. 5f). The statistical power in detecting significant co-bursting was estimated through simulation where coordinated new RNA was synthetically implanted for an increasing number of cells (Fig. 5g), demonstrating high power when co-bursting was visible in only a low percentage of cells.

Having demonstrated that the collected 2-h 4sU experiment is well suited to investigate co-bursting patterns transcriptome-wide, we focused our analysis on autosomal genes. We observed positive correlations among gene pairs when correlating new RNA profiles (irrespective

of allelic origin) (Fig. 5h, grey) and to a smaller degree when comparing allelic new RNA counts from the same allele (Fig. 5h, green), but also in the control correlations across the non-meaningful other allele (Fig. 5h, purple). The drop in correlations for allelic resolution new RNA counts is due to losing 50% of the reads compared with new RNA counts without allelic assignments. Importantly, the correlations from the same allele (in *cis*) were generally not stronger than the spurious correlations from the other allele (in *trans*), and therefore, after subtracting the median correlations (*cis*−*trans*) gene-pair correlations

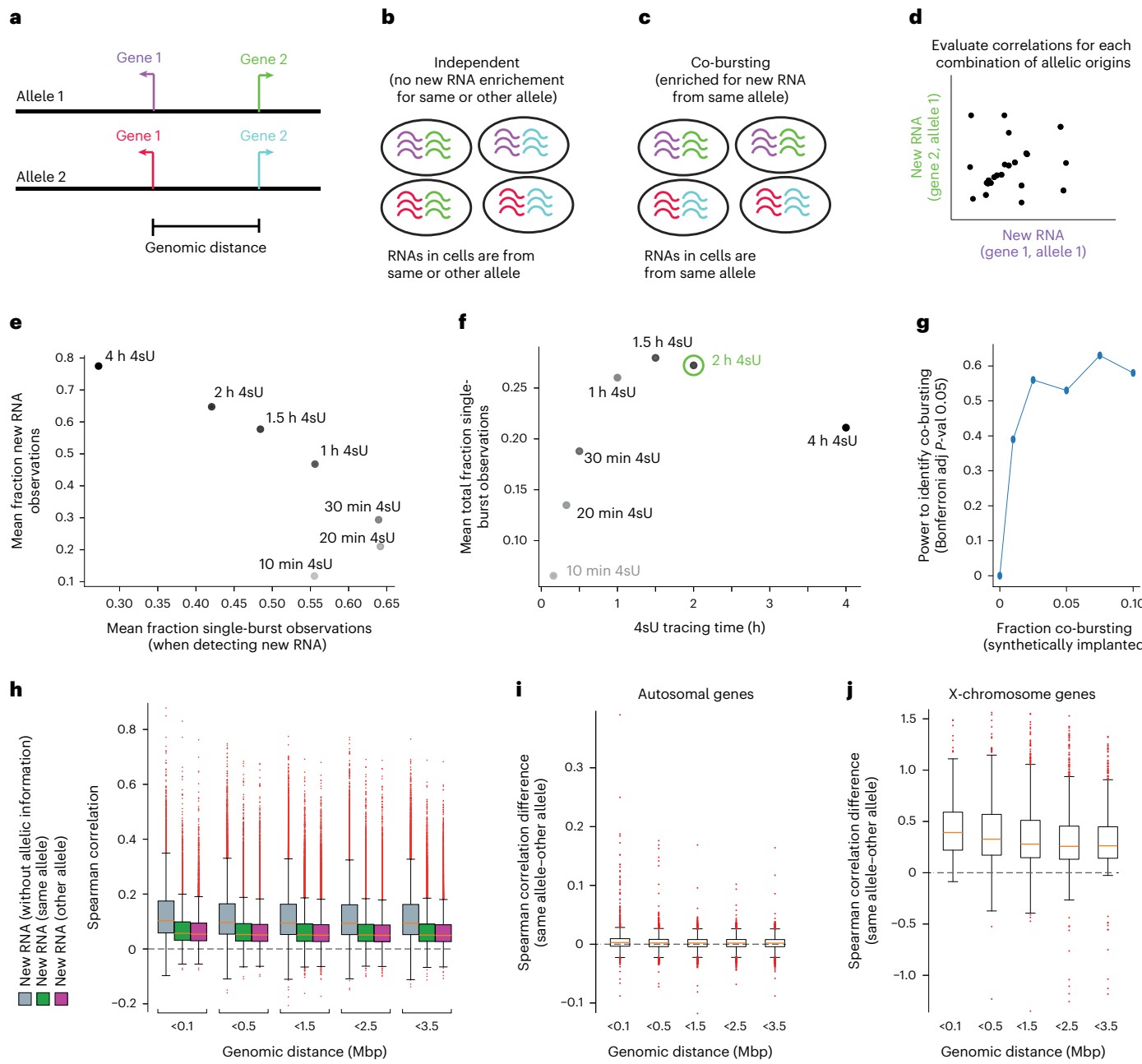

**Fig. 5 | Lack of co-busting in allelic new RNA counts from single cells.**
**a**, Illustration of a genomic region with two nearby genes. **b**, Illustration of new RNA obtained in four cells, if all alleles and genes are transcribed independently. **c**, Illustration of new RNA obtained in four cells, if nearby genes co-burst from the same allele. **d**, Illustration of a correlation of new RNA from two nearby genes on the same allele, for example from co-bursting. **e**, Simulation showing the fraction new RNA observations in cells (mean across cells; *y* axis) and the fraction new RNA observations coming from a single bursting event (*x* axis), for the indicated 4sU labelling time periods. **f**, The mean fraction of single burst new RNA observations (*y* axis) against the 4sU labelling time period. **g**, Plot showing the power to detect significant co-bursting (*y* axis) as a function of synthetically implanting coordinated new RNA counts in an increasing fraction of cells. **h**, Boxplots showing the observed correlations for pair-wise comparisons of genes within different genomic distance bins for autosomal genes. In grey

are new RNA count correlations (irrespective of allelic origin), and in green and purple are correlations obtained on new RNA counts from the same allele (green) and, as a control, from different alleles (purple). **i**, Boxplots showing the correlations from same allele (green in **h**) minus the correlations from different alleles (purple in **h**), for pair-wise autosomal genes separated by indicated genomic distance. **j**, Boxplots showing the correlations from same allele minus different alleles (as in **i**), for pair-wise genes on the X chromosome, separated by indicated genomic distance (280 gene pairs in <0.1; 881 in <0.5; 1,291 in <1.5; 900 in <2.5; 787 in <3.5). In **h** and **i**, the numbers of gene pairs are as follows: 11,308 for bin <0.1; 34,411 for <0.5; 64,773 for <1.5; 51,532 for <2.5; 50,494 for <3.5. In **h**–**j**, the boxplots show the median and boundaries (first and third quartile), and the whiskers denote 1.5 times the interquartile range of the box. Source numerical data are available in Source data.

approached zero, with a few outliers (Fig. 5i). Next, as a positive control, correlations among gene pairs on the X chromosome showed much greater correlations from the same allele (Fig. 5j). The strong allelic signal in *cis* for X-chromosome genes is obviously stemming from the

silencing of one chromosome and not from co-bursting. Finally, we investigated in more detail the autosomal gene outliers that showed sign of co-bursting (Fig. 5i, Extended Data Fig. 7 and Supplementary Table 1). The handful of outlier gene pairs contained false positives,

including multiple annotated pseudogenes expressed in one transcript (Gsm49257–Gsm18787) and imprinted gene pairs (Rian-Meg3; Extended Data Fig. 7c). Remaining gene pairs were paralogue gene pairs (for example, genes encoding for granzyme D proteins; Supplementary Table 1) or overlapping RNA on opposite strands (Extended Data Fig. 7e). We performed similar analyses using different metrics or statistical tests for detecting co-bursting at the allelic resolution, which identified sporadic paralogue gene pairs with co-busting, but no analysis provided evidence for co-bursting having a larger role in shaping the transcriptional dynamics in mammalian cells.

Surprisingly, we found no distance dependence among the new RNA correlations (Fig. 5h) despite such correlations having been previously reported[26]. The lack of general co-bursting, however, argues that such correlations should not exist. We suspected the cellular heterogeneity and batch effects present in previous data could have confounded the correlations. We tested this idea by selecting 20% of primary fibroblast cells that were the most different from the main population of cells, which created heterogeneity-derived distance-dependent correlation between genes (Extended Data Fig. 8).

## Discussion

In this study, we demonstrate that the inference of transcriptional bursting parameters can be considerably improved when analysing newly transcribed RNAs across thousands of individual cells. This was achieved through improved 4sU-based scRNA-seq (NASC-seq2) and by developing the computational inference from pulse-labelled RNA distributions. The two-state model of transcription was extended to this transient case to model the numbers of new RNAs per cell as a function of the labelling time and kinetic parameters. Noteworthily, the computational complexity of inference from the transient (pulse-labelled) state is more difficult than inference from steady-state counts, which required a dedicated inference strategy and the use of C libraries that handle computing with higher numerical precision. Profiling of nearly 10,000 fibroblast cells allowed the inference of the transcriptional off rate ($k_{off}$) and synthesis rate ($k_{syn}$) for thousands of endogenous genes, beyond the inference possible from steady-state measurements that failed to separate these parameters[9].

Measuring $k_{off}$ and $k_{syn}$ rates for thousands of endogenous genes revealed that the synthesis rate ($k_{syn}$) controls the burst size, which we demonstrated both in mouse primary fibroblasts and in human K562 cells. In contrast, the $k_{off}$ rate was revealed to be relatively constant across all genes. This was apparent since $k_{off}$ values inferred from the two data halves had less correlation (Fig. 2j), showed moderate variation (Fig. 2d) despite having similar technical noise as $k_{syn}$, and $k_{off}$ values did not correlate with burst sizes (Fig. 3c). This result agrees with a previous study that imaged nascent RNA across selected genes and similarly identified the $k_{off}$ rates to be invariant[27]. Thus, the duration of active bursting periods seems relatively constant across genes, on the order of a minute, whereas the amount of transcribed RNAs in the burst is achieved by higher synthesis rate of RNA. We found the synthesis rates to span from 3 to 200 molecules per hour (Fig. 2d). Previously, we reported that burst size of genes are higher when they contain specific core promoter elements[9]. The factors that bind the core promoter elements therefore must be able to recruit associated factors and RNA polymerases more efficiently, resulting in higher RNA transcription per burst, since the synthesis rate is regulated (as opposed to the time in the on state). As previously reported[5,9], increased burst size is predominantly used to increase the expression of the highly expressed genes, whereas modulating burst frequencies is predominantly used to tune expression for most other genes. This pattern was also apparent in our study, and importantly, the biological results of this study (that $k_{syn}$ controls burst size, whereas $k_{off}$ is invariant) applies to both the highly expressed and more moderately expressed genes (Figs. 3b–d and 4d). It will be intriguing to associate specific regulators and processes to each kinetic parameter[11], and in this pursuit, we believe single-cell new

RNA profiling (as shown with NASC-seq2) will have critical importance, in particular since new RNA profiling has the power to identify RNA effects after the perturbation of specific regulators[28], such that the extension to the single-cell level should be able to identify bursting kinetic alterations from perturbations.

The extent of bursting or constitutive expression has been debated, with large-scale experiments in bacteria[12] and human[13] that favoured predominantly constitutive expression, whereas time-lapse fluorescence microscopy at the protein level favoured predominantly transcriptional bursting[5]. Several analyses of specific genes have reported transcriptional bursting[2,4,6,7], and steady-state RNA counts from scRNA-seq better fit the model of transcription that allow for bursting[9]. Importantly, in this study, we inferred transcriptional on and off rates for thousands of endogenous RNA polymerase II transcribed genes. All genes with inferable kinetics were found to be expressed in bursts, with $k_{off}$ values typically 100-fold larger than $k_{on}$ values (Fig. 3a), providing direct RNA-level evidence for general bursting of mouse genes.

Another outstanding question is to what extent nearby genes may have coordinated bursting, so-called co-bursting, or spatial coupling of bursting. Since it is well known that nearby genes are more often involved in similar functions[29] and that the chromosomes are organized in topological domains[30], it follows naturally that nearby genes often show correlated expression across cell types[29]. However, co-bursting is, by definition, different from co-expression, and to what extent nearby genes may have coordinated bursting is debated. Reports of transcriptional hubs[15,16] and condensates[17] indirectly imply co-bursting, and co-bursting was recently observed in fruit flies by real-time imaging of transgenes[4] and paralogues[19]. Spatial coupling of bursting events was also recently reported when analysing nascent RNA[18]. The single-cell new RNA profiling across nearly 10,000 fibroblasts at allelic resolution enabled us to investigate this question transcriptome-wide. Importantly, full-length scRNA-seq methods can easily monitor allelic expression from the detection of transcribed single-nucleotide polymorphisms within sequenced reads, often employing crosses of genetically distant mouse strains (here CAST and C57Bl/6)[23,25]. We investigated whether new RNA counts of pairs of genes were more often found from the same allele (for example, both CAST) compared with cells with pairs of new RNA counts from the other allele (for example, one CAST and one C57). No such pattern was present in the data; instead, most genes have similar cell counts from the same and different alleles, indicative of independent transcriptional processes with occasional co-bursting happening by chance. The few outliers we detected were mostly technical, for example, two imprinted genes or two pseudogenes expressed from the same RNA transcript. We did see examples of paralogues with higher proportion of cells with nascent RNA from the same alleles, possibly indicating that certain paralogue gene pairs may be more prone to co-bursting. This is in stark contrast to the two alleles of the same genes that are always statistically independent, despite most often having identical gene regulatory elements. Thus, shared gene regulatory elements (as for paralogues[19]) together with closer genomic distances, might be a prerequisite for the sparse, few examples of coordinated co-bursting in eukaryotic genomes.

It is important to note the limitations with this study. It is not fully understood to what extent the 4sU exposure and incorporation into RNA affects the cells and the library construction. Typically, the complexity in single-cell RNA-seq libraries from 4sU exposed cells are smaller than in cells unexposed to 4sU, and with new RNA profiling in single cells the burst sizes obtained seem systematically underestimated compared with inference from standard scRNA-seq[9] and other studies[10,11]. It is probably a combination of slight hindrance of 4sU-incorporated RNAs during library construction and false negative assignment of RNAs due to too sparse T-to-C conversions in the sequenced reads. Yet, with the single-cell new RNA profiling

method we developed in this study, NASC-seq2, we show that it is possible to detect over 100,000 RNA molecules per cell (Fig. 1f) for sensitive analyses of transcriptional dynamics at the resolution of individual bursts.

## Online content

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

## Methods

### Primary fibroblast derivation

Primary mouse fibroblasts were obtained from the tail of a 5-month-old female CAST/EiJ × C57BL/6J mouse (ethical permit numbers N95/15 and 13572-2020 from Jordbruksverket) by removal of the tail skin and culturing the minced tail in Dulbecco's modified Eagle medium (high glucose, Gibco) supplemented with 10% embryonic stem cell foetal bovine serum (FBS; Gibco), 1% penicillin–streptomycin (Gibco), 1% non-essential amino acids (Gibco), 1% sodium pyruvate (Gibco) and 0.1 mM β-mercaptoethanol (Gibco) in a humidified incubator at 37 °C and 5% $CO_2$. Tail explants were removed after 5 days, and the cultures were passaged twice to obtain a fibroblast culture. Cells were then frozen and stored in 90% FBS and 10% DMSO until needed for the experiment.

### Fibroblast cell culture and labelling

Cells were thawed and passaged twice and cells were seeded 10,000 cells per well of a six-well plate. The next day, 4-thiouridine (4sU; Sigma) was added to the growth medium to a final concentration of 200 μM. No 4sU was added to the wells that served as unlabelled control. After 2 h of labelling time, cells were detached using trpLE (Gibco), washed in cold Dulbecco's phosphate-buffered saline and transferred through a mesh (35 μm, Corning), keeping the cell suspension on ice until sorting. We performed four experiments as described, after which cells were pooled for fluorescence-activated cell sorting (FACS) and downstream analyses.

### K562 cell culture and labelling

K562 cells were obtained from DSMZ and authenticated by DSMZ Identification Service according to standards for STR profiling (ASN-0002). K562 cells (100,000) cultured in RPMI (Gibco) supplemented with 10% FBS (Sigma-Aldrich), 1× GlutaMAX supplement (Gibco) and 1% penicillin–streptomycin (Gibco) were seeded into each well of a six-well plate on the day before the experiment. Cells were labelled for 2 h with 200 μM 4sU, washed and resuspended in cold Dulbecco's phosphate-buffered saline and transferred through a 35 μm mesh and kept on ice until sorting.

### Preparation of the 4sU-containing spUMI pool

A 5′ molecular spike plasmid pool[31] was in vitro transcribed using a T7 MaxiScript (Thermo Fisher Scientific) according to the manufacturer's protocol, with 10% of dUTP replaced by 4sUTP (Jena Bioscience). The resulting spUMI was purified by RNeasy column (Qiagen).

### Cell sorting

Cells were sorted on a BD FACS Melody cell sorter (BD Biosciences). Propidium iodine (Invitrogen) was used to stain for dead cells and the general gating strategy during FACS is shown in Extended Data Fig. 1i. From the live population, single cells were sorted through a 100 μm nozzle into 384-well polymerase chain reaction (PCR) plates with 0.3 μl lysis buffer containing 2.5 U μl$^{-1}$ recombinant RNase inhibitor (RRI, 40 U μl$^{-1}$, Takara), 0.04 pg μl$^{-1}$ of the aforementioned 4sU-containing spUMI pool and 0.1% Triton X-100, overlaid by 3 μl Vapor-Lock (Qiagen) per well. Plates were sealed, spun down immediately after sorting and stored at −80 °C until further processing.

### NASC-seq2 sequencing library preparation

Plates were taken from −80 °C, briefly spun down and kept on ice until adding 0.3 μl of alkylation mix containing 50 mM Tris–HCl (pH 8.4), 10 mM iodoacetamide (Sigma-Aldrich, dissolved in DMSO) and a total percentage of 45% DMSO (concentrations calculated for 0.6 μl alkylation volume). Plates were alkylated for 15 min at 50 °C and quenched with 0.4 μl of quenching mix containing 35 mM dithiothreitol (DTT, Sigma), 0.5 mM dNTPs, 0.6 μM oligo-dT primer (5′-biotin-ACGAGCATCAGCAGCATACGATTTTTTTTTTTTTTTTTTTTTTTTTTTTTTTVN-3′; IDT) and 0.4 U μl$^{-1}$ RRI for 5 min at room temperature followed by a denaturation step at 72 °C for 10 min. Concentrations were calculated for 1 μl final quenching volume (DTT) and 4 μl total reaction volume of RT (dNTPs, oligo-dT primer, RRI), respectively. For individual molecule counting, NASC-seq2 uses the UMI-containing Smart-seq3 template-switching oligo[32] (5′-biotin-AGAGACAGATTGCGCAATGNNNNNNNNrGrGrG-3′; IDT) in the RT reaction. A mix of 25 mM Tris–HCl (pH 8.0), 35 mM NaCl, 1 mM GTP (Tris-buffered, Thermo Fisher Scientific), 2.5 mM $MgCl_2$, 5% Polyethylene Glycol (PEG), 2 mM DTT, 0.4 U μl$^{-1}$ RRI, 2 μM template-switching oligo and 2 U μl$^{-1}$ Maxima H-minus reverse transcriptase (Thermo Fisher Scientific) was prepared with indicated concentrations applying to the final RT volume of 4 μl. The dilution of DMSO from 45% in the alkylation reaction to below 7% in the RT reaction allows for the use of Maxima H-minus RT enzyme, which is sensitive to high concentrations of DMSO. Three microlitres of RT mix was dispensed into each well and incubated at 42 °C for 90 min, 10 cycles of 50 °C and 42 °C for 2 min each, and final denaturation for 5 min at 85 °C. The remaining library preparation follows the smart-seq3 protocol[32].

Briefly, 6 μl pre-amplification PCR mix was added per well containing 1× KAPA HiFi PCR buffer (2 mM Mg at 1×, Roche), 0.02 U μl$^{-1}$ KAPA HiFi HotStart DNA polymerase (Roche), 0.5 μM forward primer (5′-TCGTCGGCAGCGTCAGATGTGTATAAGAGACAGATTGCGCAATG-3′; IDT) and 0.1 μM reverse primer (5′-ACGAGCATCAGCAGCATACGA-3′; IDT), 0.3 mM dNTPs and 0.5 mM $MgCl_2$. PCR was performed at 98 °C for 3 min, 21 cycles of 98 °C for 20 s, 65 °C for 30 s and 72 °C for 4 min, followed by 5 min final extension at 72 °C.

Amplified cDNA was cleaned up with 6 μl of home-made SPRI beads in 22% PEG and eluted in 12 μl ultrapure water (Invitrogen). Per-well cDNA concentrations were quantified using the QuantiFluor dsDNA assay (Promega) on an FLUOstar Omega plate reader (BMG Labtech) and diluted with ultrapure water to 200 pg μl$^{-1}$.

One microlitre of diluted cDNA was tagmented with 1 μl tagmentation mix containing TD buffer (final concentration 10 mM Tris–HCl (pH 7.5), 5 mM $MgCl_2$ and 5% N,N-dimethylformamide) and 0.08 μl Tn5 enzyme (ATM, Illumina Nextera XT sample preparation kit) per well for 10 min at 55 °C. Next, 0.5 μl of freshly prepared 0.2% sodium dodecyl sulfate was added and incubated for 5 min at room temperature.

For tagmentation PCR, 1.5 μl of custom Nextera index primers (0.5 μM each) was added per well followed by the addition of 3 μl tagmentation PCR mix with concentrations of 1× Phusion HF buffer (Thermo Fisher Scientific), 0.2 mM dNTPs and 0.01 U μl$^{-1}$ Phusion DNA polymerase (Thermo Fisher Scientific) in the final PCR volume of 7 μl. Tagmentation PCR was performed at 72 °C for 3 min for gap filling, initial denaturation at 98 °C for 3 min, followed by 12 cycles of 98 °C for 10 s, 55 °C for 30 s and 72 °C for 30 s, with final elongation of 5 min at 72 °C. Final indexed library was pooled and purified using 0.6× volume home-made SPRI beads in 22% PEG. Library concentrations were quantified using the dsDNA Qubit kit (Invitrogen) and visualized using the Agilent bioanalyzer high-sensitivity DNA kit. The full protocol of NASC-seq2 has also been deposited in protocols.io (https://doi.org/10.17504/protocols.io.6qpvr43nogmk/v1).

### Library circularization and MGI sequencing

Libraries were pooled and converted into circular libraries through five cycles of adapter conversion PCR and circularization of 1 pmol of the product using the MGIEasy Universal Library Conversion kit App-A (MGI Tech.). Final circular, single-stranded DNA (ssDNA) sequencing libraries were quantified with the ssDNA Qubit kit (Invitrogen). DNA nanoballs were made from the circular ssDNA library pools using a custom primer (5′-TCGCCGTATCATTCAAGCAGAAGACG-3′) for rolling-circle amplification and loaded onto Flow Cell Large (FCL) flow cells (MGI Tech.). Custom sequencing primers were added to the sequencing cartridges ((read 1: 5′-TCGTCGGCAGCGTCAGATGTGTATAAGAGACAG-3′; MDA: 5′-CGTATGCCGTCTTCTGCTTGAATGATACGGCGAC-3′,

read 2: 5′-GTCTCGTGGGCTCGGAGATGTGTATAAGAGACAG-3′; i7 index: 5′-CCGTATCATTCAAGCAGAAGACGGCATACGAGAT-3′; i5 index: 5′-CTGTCTCTTATACACATCTGACGCTGCCGACGA-3′)). The resulting libraries were sequenced on an MGI DNBseq G400RS using StandardMPS PE200 reagents.

### Processing of NASC-seq2 sequence data

zUMIs[33] (v. 2.9.7) was used to process raw FASTQ files. First, reads were filtered on the basis of the cell barcode quality (five bases with a Phred score <20). The 5′ UMI containing reads were identified by the pattern ATTGCGCAATG with up to two mismatches. UMI containing reads with a poor-quality UMI sequence were filtered out (three bases with a Phred score <20). Reads were mapped to the human (hg38) or mouse (mm39) genome using STAR (v. 2.7.1 for human and v. 2.7.3a for mouse respectively) and error corrected UMI counts were quantified based on gene annotations (ENSEMBL GRCh38.95 for human and GENCODE GRCm39.vM29 for mouse, respectively). Previously published K562 NASC-seq cells[20] were re-processed as above to have identical processing and gene annotations to the K562 NASC-seq2 data.

### Reconstructing RNA molecules from paired reads sharing 5′ UMI sequence

Partial reconstruction of RNA molecules up to 1 kb is feasible from paired-end short-read sequencing data where the 5′ end contain an UMI[32]. Since the ability to separate new and old RNA is highly dependent on the RNA sequence length (Extended Data Fig. 1a), reconstructing RNA sequences longer than that obtained from the paired reads alone, benefit the RNA classification. To this end, we used the UMI-containing reads and reconstructed their other paired reads using stitcher.py (https://github.com/AntonJMLarsson/stitcher.py). Briefly, paired-end reads with the same error-corrected UMI sequences were grouped, and the merged reconstructed sequence was written to a new bam file. The Phred scores of the merged sequences was propagated in cases more than one read sequence covered a base. The likelihood of each base call was derived from the Phred score, and the remaining likelihood of the other three bases being correct was distributed equally. If the initial sequencer base call as N, then the likelihood was distributed equally over all four bases. The most likely base call for each position was calculated using the softmax function, and probability scores above 0.3 were considered sufficient for a A, T, C or G base call. If the probability was below 0.3, the base call became N. The corresponding Phred score was calculated as $-10 \times \log_{10}(1 - p\_max)$.

### Filter mismatches to remove sequence errors and genetic polymorphism

Reconstructed molecules were compared with the reference genome to extract the number of mismatches. Depending on gene strand, the position of each T > C or A > G mismatch was saved. Mismatches were evaluated using a binomial distribution $B(k|n, p)$, where $k$ is the number of molecules with the observed mismatch in that position, $n$ is the number of molecules that cover the position and $p$ is the median fraction of mismatches over all positions. All positions considered significant ($\alpha = 0.05$ after Bonferroni correction) were masked.

### Estimating probability of conversion and assigning RNA molecules as new

To estimate the probability of observing a converted position (T > C for positively stranded genes and A > G for negatively stranded genes) we applied the previously described expectation-maximization algorithm[20,34]. Briefly, the mismatch statistics are gathered for each reconstructed molecule for each cell. We consider the mismatches to arise from a two-component binomial mixture, where one component is governed by the conversion probability ($p_c$) and the other component is governed by the error probability ($p_e$). To estimate the error probability, we used the mean of the C > T and G > A mismatch rates.

The statistics for the mismatch expected in new molecules (T > C or A > G) and the $p_e$ were used in the expectation-maximization algorithm to obtain the $p_c$ estimate. For molecule-level hypothesis testing, we used the likelihood-ratio test with the null hypothesis $H_0: p = p_c$ and alternative hypothesis $H_A: p = p_c$ with a binomial likelihood at $\alpha = 0.05$. Each molecule was then genotyped according to the observed single-nucleotide variants that have been validated to segregate the two mouse strains, only genotyping the molecule as maternal or paternal if exclusively maternal or paternal variants were observed.

### Derivation of pulse-labelled RNA probability max function for the telegraph model

Using the steady-state probability generating function for the telegraph model, we used common tools used in mathematical analysis to derive the probability distribution of observing $n$ molecules after labelling time $t$. Due to the numerical instability of evaluating Kummer's function in Python libraries (for example, scipy) that are crucial to this computation, we implemented this computation using the C library Arb[35], which allows for arbitrary precision of variables and also contains a very accurate module to compute Kummer's function. More information on the derivation of the new RNA probability mass function is available in Supplementary Note 1, and the Arb implementation is available on GitHub (see 'Code availability' statement).

### Sample quality filter

Based on the valley of the bimodal distribution of reconstructed molecules per sample, K562 cells were filtered on the basis of having over 2,700. For fibroblasts, we required over 4,000. This removal of low-quality samples caused suspiciously artefact-like bimodal distributions in the kinetic estimates to resolve and disappear.

### Inference of transcriptional bursting from new RNA profiling of individual cells

As the function in Supplementary Note 1 takes kinetic parameters and outputs the probability distribution of observing $n$ molecules, and our aim is to go the opposite direction, from molecular count data to the inference of kinetic parameters, we devised a new strategy to numerically invert the function. In this three-step strategy, a look-up table was first created and used to approximate kinetic parameters from count summary statistics, used as parameter initialization in the subsequent maximum likelihood inference. To create a look-up table, we first simulated new RNA counts for combinations of kinetic parameters. The new RNA count vectors obtained for each kinetic parameter combination was summarized using three summary statistics (fraction cells with new RNA expression, average expression among cells with expression, coefficient of variation in new RNA counts among cells with expression). The kinetic parameters used for modelling together with the summary statistics were turned into a look-up table. The simulations spanned 73 different $k_{on}$ values (from 0.002 to 50), 38 different $k_{syn}$ values (1 to 200) and 55 different $k_{off}$ values (0.25 to 500). The values were chosen equidistant on log scale, so that they are steps of ~1.15×. We chose these boundaries because they covered the data's distribution and the limit on $k_{syn}$ and $k_{off}$ size was needed as larger values resulted in bimodal errors in simulations. The 2-h 4sU incubation period was set, and the gene-level degradation rates were inferred as 0.065 h⁻¹ (calculated as −ln(fraction_old)/time using fibroblasts). To avoid numerical error propagations, we used 10,000-bit floating point numbers for the formula in Supplementary Note 1. Where possible when performing look-up searches, we used linear interpolation to identify initial kinetic parameters, and when linear interpolation failed, we simply identified the closest trio of kinetic parameters. Due to limitations in the linear interpolation, only summary values corresponding to the expression interval 0.01–350 counts per cell were used.

Next, we calculated the same three summary statistics and used them in conjunction with the look-up table to interpolate kinetic parameters or, where interpolation broke down, take the closest entry

(in terms of summary statistics) in the look-up table. If identified kinetic parameters were on the boundary, the gene was excluded. Bootstrapping was used on the initial cells to infer confidence intervals to each parameter (50 bootstraps per gene), and we denoted robust inferences (or 'controlled' inferences) when the difference between upper and lower quartile was below 2 and when less than 50% of the bootstrap inferences failed to identify parameters.

The kinetic parameters identified from the look-up table was next used as initialization for maximum likelihood estimation to estimate the kinetic parameters: $k_{on}$, $k_{syn}$ and $k_{off}$. This uses a log likelihood function for match between the observed count distribution, and formula's probability distribution of observing $n$ molecules given kinetic parameters. An optimization algorithm (L-BFGS-B) search for the kinetic parameters that maximized likelihood.

Based on the kinetic estimates, we also calculated mean occupancy (fraction of time in on state) = $k_{on}/(k_{on} + k_{off})$, burst size (RNAs per on state) = $k_{syn}/k_{off}$, burst frequency (on states per time) = $1/(1/k_{on} + 1/k_{off})$, and expression rate (RNAs per time) = $k_{syn} \times k_{on}/(k_{on} + k_{off})$.

### Standard deviations
For figures where the data were split in halves, the geometric standard deviation was calculated as exp(sqrt(avg((log(half1)-log(half2))$^2$ for genes)/2)). For figures without that split, it was calculated as geom_avg(95%CI_high/95%CI_low for genes)$^{(1/(2 \times 1.96))}$ on the basis of bootstrapped confidence intervals.

### Simulating the bias in inference from pulse-labelled and steady-state RNA count distributions
Through simulation, we created count tables of 4,000 cells from the distributions of the formula in Supplementary Note 1 using $k_{on}$ values of 0.2, 0.8944 and 4, $k_{syn}$ 10, 22.36 and 50 and $k_{off}$ 30, 77.46 and 200, corresponding to low, middle and high values of their distributions, with 100 simulations per each value combination, degradation at 0.065 and time either 2 h (for nascent RNA) or 1,000 h (for pre-existing + nascent RNA). We then estimated parameters from each simulated data set and plotted the deviation of inferred parameters from the input (true) values of $k_{on}$, $k_{syn}$ and $k_{off}$. Estimation of kinetic parameters from total (pre-existing + new) RNA was done as previously described[9].

### Read-level quantification of temporal state and allelic origin
Read-level quantification of temporal state and allelic origin was performed for the co-bursting analysis (Fig. 5), although similar results were reached while using molecule-level assignment of temporal state and allelic origin. For each read pair (all sequencing was performed using PE200), we assigned the read to allelic origin based on the observed single-nucleotide variants that separate the two mouse strains (CAST and C57Bl/6). Reads without transcribed genetic variation were removed. Next, using the 4sU-induced mismatches per read, we performed hypothesis testing to assign reads as new. The hypothesis testing was generated using the $p_e$ and $p_c$ estimates obtained on the molecule-level analysis above.

### Assessing pair-wise coordination of transcriptional bursting
Pairs of read-level new RNA counts per gene was compared using Spearman correlations in scipy, either when using all new RNA counts, or after stratifying RNA counts to allelic origin. Comparison between new RNA counts between two genes of the same chromosome was named *cis*, whereas non-meaningful comparisons of two gene on to different chromosomes were named *trans*. Additionally, chi-square and Fisher's exact tests (scipy) were applied to allele-assigned new RNA counts for gene pairs, with similar conclusions.

### Statistics and reproducibility
No statistical method was used to pre-determine sample sizes (that is, cell numbers) throughout the study. We aimed to create a highly informative data set on transcriptional bursting in one cell-type with both high cell numbers (close to 10,000) and deep RNA counting per cell (median 100,000) to have sufficient power to infer transcriptional bursting and co-bursting, since power increase with cell numbers (Fig. 2k). The experiments were not randomized, and the investigators were not blinded to allocation during the experiments and outcome assessment. The NASC-seq2 data set from K562 cells was used for Fig. 1a and Extended Data Figs. 1 and 4 and were based on one biological experiment involving 613 post-quality-control filtered cells, where data for each cell were treated independently throughout NASC-seq2 library preparation and analyses. The improvement with NASC-seq2 over NASC-seq has been repeatedly observed in K562 cells and primary fibroblasts. The large-scale NASC-seq2 experiment on F1 primary fibroblasts was generated from four technical replicates of primary fibroblast cultures that were independently exposed to 4sU and collected for FACS and NASC-seq2 library construction; each cell was treated independently throughout NASC-seq2 library preparation and initial analyses. For transcriptional dynamics and co-bursting analyses of the primary fibroblasts (Figs. 1–5 and Extended Data Figs. 2, 3 and 5–8), cells from all replicates were pooled before analyses since as they had uniform transcriptional patterns (Fig. 1c). Inferences of transcriptional bursting parameters was performed in parallel on independent subset of cells to avoid spurious correlations among inferred parameters (as described in detail in Methods). Throughout the study, $P$ values refer to two-tailed tests, with the details on the statistical analysis performed for each type of data analysis reported in the respective Methods section. Spearman correlation analyses (in Figs. 3d and 4d and Extended Data Fig. 4d) were performed in the analysis since only real positive numbers were possible, including zeros, and since the parameters did not easily follow a normal or lognormal distribution. For Extended Data Fig. 8, analyses used a binomial test to assess if the median was significantly departed from zero (as it allows the null hypothesis that correlations are distributed above or below zero with equal probability). In Extended Data Fig. 3b,c, we use Hartigan's dip test for unimodality because the data seem to follow a normal distribution (Extended Data Fig. 3a).

### Reporting summary
Further information on research design is available in the Nature Portfolio Reporting Summary linked to this article.

### Data availability
Raw NASC-seq2 sequencing data (K562 and primary fibroblast cells) have been deposited in ENA (accession ID: PRJEB60799) and source data has been deposited in Zenodo (https://doi.org/10.5281/zenodo.12092003). Kinetic estimates, code and count tables are available on GitHub (https://github.com/sandberg-lab/NASC-seq2). We downloaded genome sequences from UCSC Genome Browser (mouse: GRCm39/mm39 and human:GRCh38/hg38) and GENCODE gene annotations (human ENSEMBL GRCh38.95 and mouse GRCm39.vM29). Source data are provided with this paper.

### Code availability
We have deposited code for processing and analyses of NASC-seq2 data on GitHub (https://github.com/sandberg-lab/NASC-seq2).

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

## Acknowledgements

This work was supported by grants to R.S. from the Swedish Research Council, the Knut and Alice Wallenberg Foundation, Karolinska Institutet, the Göran Gustafsson Foundation and the Torsten Söderberg Foundation.

## Author contributions

R.S., G.-J.H. and A.J.M.L. conceived the overall study. G.-J.H. and M.H.-J. developed the NASC-seq2 method. G.-J.H. and J.V.M. performed the NASC-seq2 experiments of the study. A.J.M.L., C.Z. and L.H. developed the computational processing of sequence data and assignments to temporal and allelic states. A.J.M.L. derived the new RNA probability mass function and original implementations. D.R., A.J.M.L. and R.S. performed all biological analyses of bursting kinetics from NASC-seq2 data. R.S. wrote the manuscript, with input from A.J.M.L., D.R. and G.-J.H.

## Funding

## Competing interests

The authors declare no competing interests.

## Additional information

**Extended data** is available for this paper at https://doi.org/10.1038/s41556-024-01486-9.

**Correspondence and requests for materials** should be addressed to Rickard Sandberg.

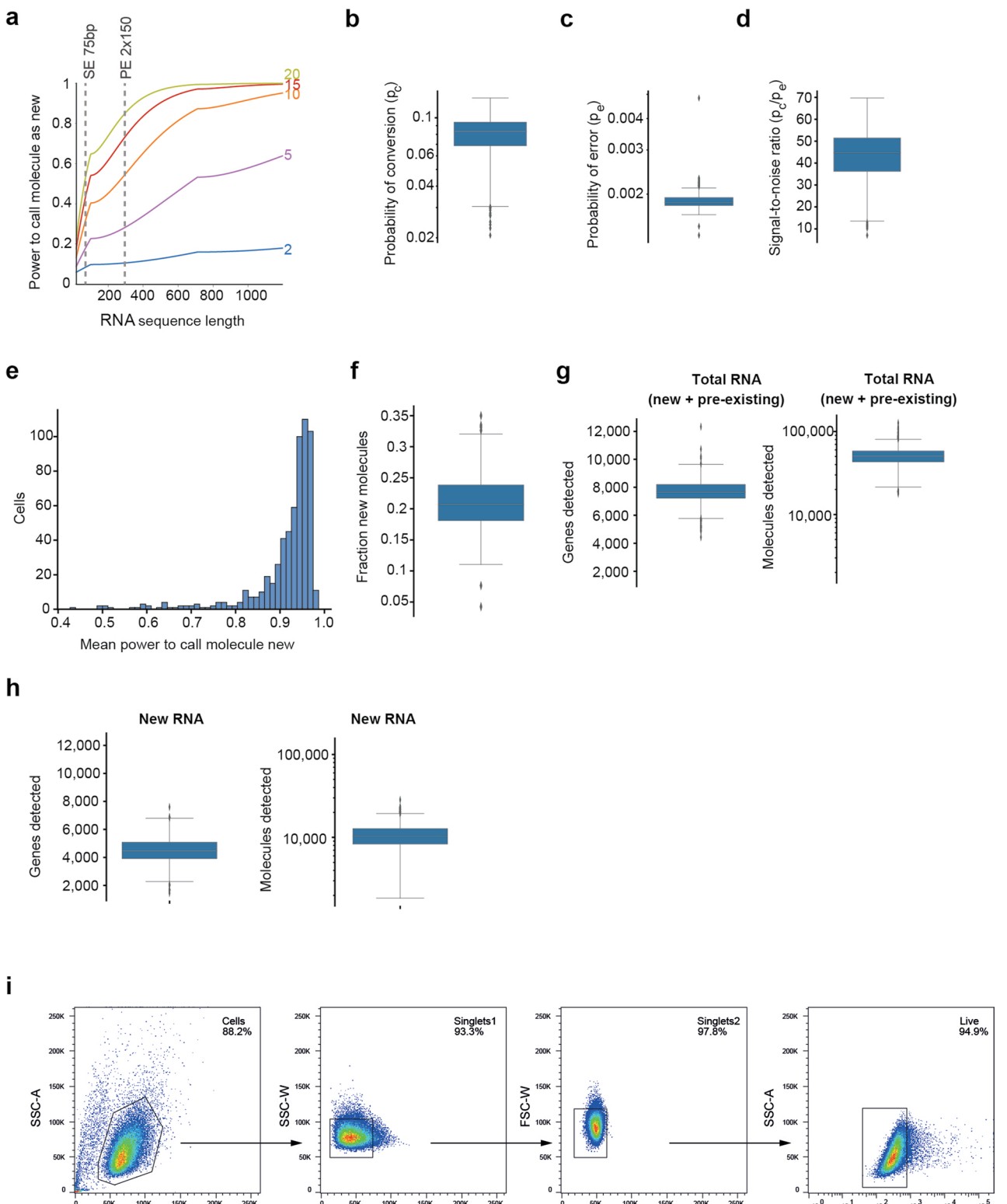

**Extended Data Fig. 1 | NASC-seq2 quality control data for K562 cells. (a)** Line plots showing the power to assign RNA molecules as new, as a function of the reconstructed sequence length, colored according to different 4sU-induced conversion levels (signal-to-noise: Pc/Pe). Dashed vertical lines show the power obtained without molecule-level reconstruction for single-end 75bp and paired-end 150bp sequencing, respectively. **(b)** Boxplot showing the inferred probability of conversion (Pc) in K562 cells (n = 613). **(c)** Boxplot showing the inferred probability of error (Pe) in K562 cells (n = 613). **(d)** Boxplot showing the signal-to-noise in K562 cells (n = 613). **(e)** Histogram summarizing the number of molecules and the power to assign them as new. **(f)** Boxplot showing the fraction of RNA molecules assigned as new in K562 cells. **(g)** Boxplots with number of genes (left) and RNA molecules (right) detected based on all RNAs (new and pre-existing). **(h)** Boxplots with number of genes (left) and RNA molecules (right) detected in new RNA. Source numerical data are available in source data. **(i)** Gating strategy for single-cell sorting into 384-well plates. Initial gating selects for cells and in particular singlets (avoiding doublets), whereas gating for Propidium Iodide selects for live cells.

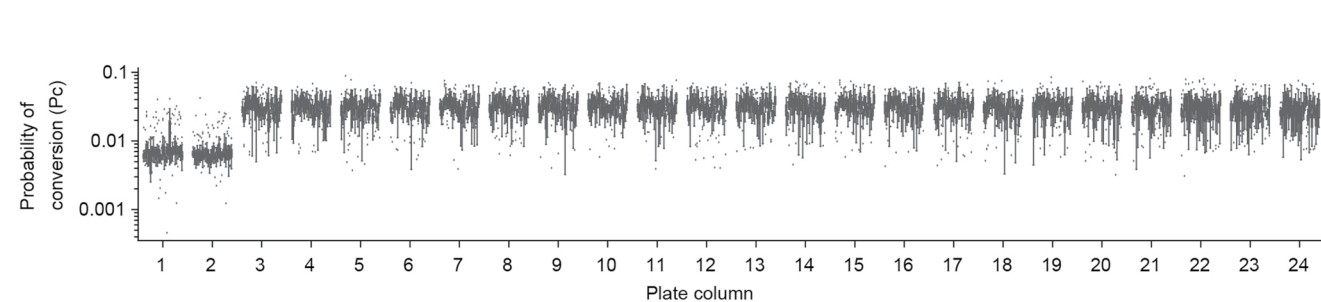

**Extended Data Fig. 2 | See next page for caption.**

**Extended Data Fig. 2 | NASC-seq2 quality control data for primary fibroblasts.** (**a**) Histogram with total number of genes detected using all RNA molecules (new+pre-existing), for cells with 4sU (blue) and controls cells without 4sU (yellow). (**b**) Histogram with total number of RNA molecules (new+pre-existing), for cells with 4sU (blue) and controls cells without 4sU (yellow). (**c**) Histogram with fraction new RNA, for cells with 4sU (blue) and controls cells without 4sU (yellow). (**d**) Boxplot showing the inferred probability of conversion (Pc) in primary fibroblast data. (**e**) Boxplot showing the inferred probability of error (Pe) in primary fibroblast data. (**f**) Boxplots showing the probability of error across all primary fibroblasts cells, stratified by plate column of cells. (**g**) Boxplots showing the probability of conversion across all primary fibroblasts cells, stratified by plate column of cells. Note that cells in columns 1 and 2 on each plate of primary fibroblasts where control cells that were not exposed to 4sU. Source numerical data are available in source data.

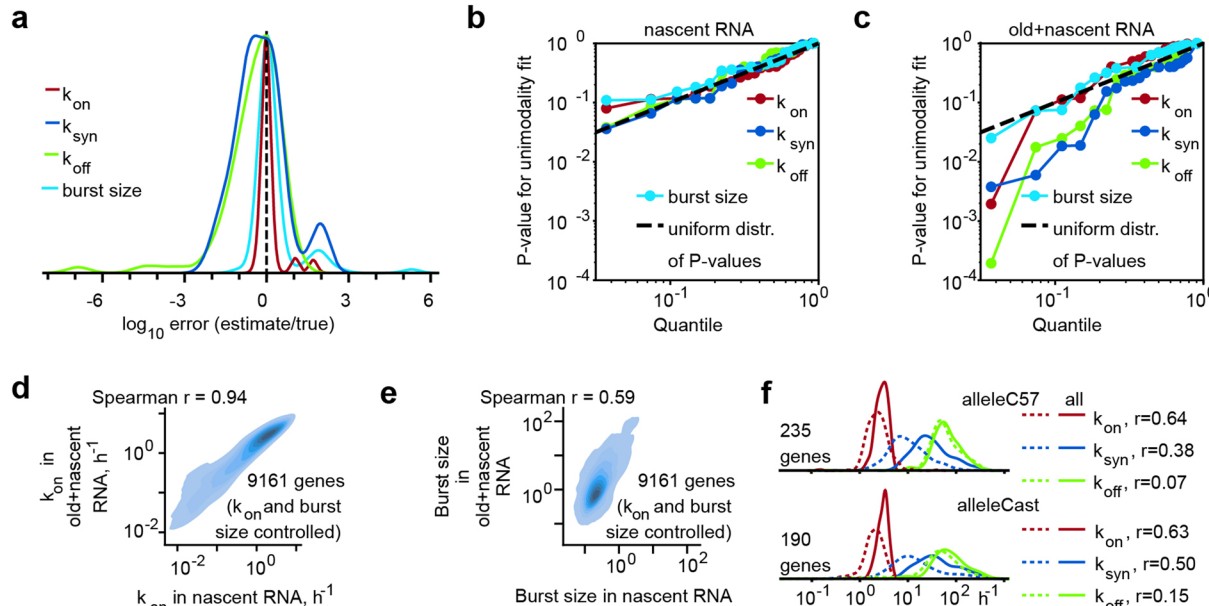

**Extended Data Fig. 3 | Assessing inference accuracy and bias through simulation.** (**a**) Density plots showing the error in inferred parameter compared to the true input to the simulation. For each parameter, we simulated 4000 individual cells for 27 combinations of *kon*, *ksyn* and *koff* values (corresponding to the low, middle and high end of their distributions), and repeated each combination 100 times to obtain statistics on the inference error and bias. Data in a-c was based on 27 independent simulations. (**b**) Hartigans' dip test for fit to unimodality for the three error distributions (from a) and burst size. All were found to be unimodal. (**c**) Hartigans' dip test for fit to unimodality for the three error distributions (from a) and burst size, when instead inferred from steady-state RNA counts (that is new + pre-existing RNA). Here, *koff* and *ksyn* obtain bimodal error distributions, giving rise to a false bimodal distribution

in parameters when not inferred from nascent RNA. (**d**) Contour plots showing correlations between burst frequency inferred from new RNA counts (x-axis) against burst frequency inferred from total RNA counts using the steady-state inference model (y-axis). (**e**) Contour plots showing correlations between burst size inferred from new RNA counts (x-axis) against burst size inferred from total RNA counts using the steady-state inference model (y-axis). (**f**) Estimation from allelic new RNA counts give half as large *kon* values, as expected, but also lower *ksyn* values from an inefficiency to assign RNA molecules to both temporal state and allelic origin (that is there are many RNA molecules that are discarded which brings down the synthesis rate). Therefore, fewer genes retain their small confidence intervals. Source numerical data are available in source data.

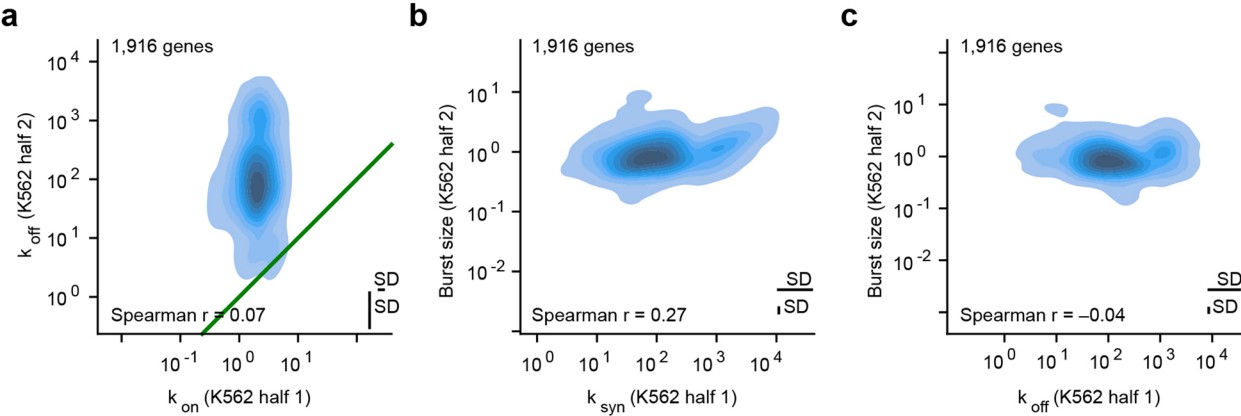

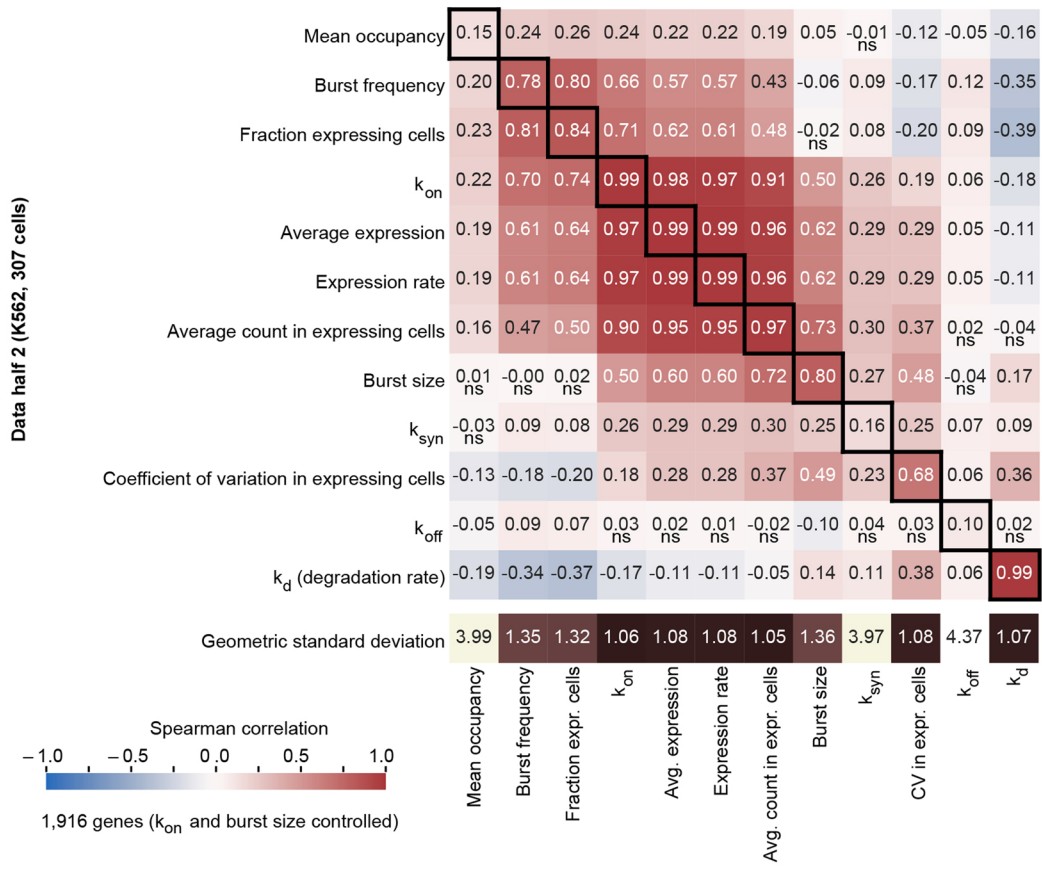

**Extended Data Fig. 4 | Transcriptional bursting analysis in the smaller human K562 NASC-seq2 data.** (**a**) Contour plot of transcriptional on (*kon*) and off (*koff*) rates, inferred separately on two non-overlapping halves of the cells (306 and 307 cells per half), across 1,916 genes with robust inference of kon and burst size. (**b**) Contour plot of burst size (inferred from cell subset, or half, 2) against the synthesis rate, *ksyn*, (inferred from cell subset, or half, 1) (1,916 genes with robust inference of kon and burst size). (**c**) Contour plot of burst size (inferred from cell subset, or half, 2) against the off rate, *koff*, (inferred from cell subset, or half, 1) (1,916 genes with robust inference of *kon* and burst size). (**d**) Spearman correlation matrix of all measurements (asterisks) and derived estimates from K562 cells, when in parallel inferred from the two cell subsets (or halves) based on the 1,916 genes with robust *kon* and burst size inference. Geometric standard deviation indicates technical variation. Source numerical data are available in source data.

**a**   Linear model for burst size

| Dep. Variable: | bs | R-squared: | 0.136 |
|---|---|---|---|
| Model: | OLS | Adj. R-squared: | 0.131 |
| Method: | Least Squares | F-statistic: | 25.90 |
| Date: | Wed, 13 Mar 2024 | Prob (F-statistic): | 1.06e-28 |
| Time: | 15:25:34 | Log-Likelihood: | -120.11 |
| No. Observations: | 995 | AIC: | 254.2 |
| Df Residuals: | 988 | BIC: | 288.5 |
| Df Model: | 6 | | |

| | coef | std err | t | P > \|t\| | [0.025 | 0.975] |
|---|---|---|---|---|---|---|
| Intercept | -0.0417 | 0.091 | -0.460 | 0.646 | -0.220 | 0.136 |
| gl | -0.0870 | 0.021 | -4.129 | 0.000 | -0.128 | -0.046 |
| EPD_INR | -0.2909 | 0.166 | -1.753 | 0.080 | -0.616 | 0.035 |
| gl:EPD_INR | 0.0789 | 0.038 | 2.076 | 0.038 | 0.004 | 0.153 |
| EPD_TATA | 1.6183 | 0.298 | 5.423 | 0.000 | 1.033 | 2.204 |
| gl:EPD_TATA | -0.3212 | 0.075 | -4.304 | 0.000 | -0.468 | -0.175 |
| EPD_TATA:EPD_INR | -0.0256 | 0.088 | -0.290 | 0.772 | -0.199 | 0.148 |

| Omnibus: | 186.479 | Durbin-Watson: | 1.948 |
|---|---|---|---|
| Prob(Omnibus): | 0.000 | Jarque-Bera (JB): | 354.147 |
| Skew: | 1.106 | Prob(JB): | 1.25e-77 |
| Kurtosis: | 4.909 | Cond. No. | 167. |

Notes:
[1] Standard Errors assume that the covariance matrix of the errors is correctly specified.
  gl = gene length

**b**   Linear model for burst frequency

| Dep. Variable: | bf | R-squared: | 0.043 |
|---|---|---|---|
| Model: | OLS | Adj. R-squared: | 0.037 |
| Method: | Least Squares | F-statistic: | 7.368 |
| Date: | Wed, 13 Mar 2024 | Prob (F-statistic): | 1.00e-07 |
| Time: | 15:25:21 | Log-Likelihood: | 482.44 |
| No. Observations: | 995 | AIC: | -950.9 |
| Df Residuals: | 988 | BIC: | -916.6 |
| Df Model: | 6 | | |

| | coef | std err | t | P > \|t\| | [0.025 | 0.975] |
|---|---|---|---|---|---|---|
| Intercept | 0.6809 | 0.050 | 13.744 | 0.000 | 0.584 | 0.778 |
| gl | -0.0697 | 0.012 | -6.059 | 0.000 | -0.092 | -0.047 |
| EPD_INR | -0.2038 | 0.091 | -2.251 | 0.025 | -0.381 | -0.026 |
| gl:EPD_INR | 0.0430 | 0.021 | 2.074 | 0.038 | 0.002 | 0.084 |
| EPD_TATA | -0.3163 | 0.163 | -1.942 | 0.052 | -0.636 | 0.003 |
| gl:EPD_TATA | 0.0779 | 0.041 | 1.912 | 0.056 | -0.002 | 0.158 |
| EPD_TATA:EPD_INR | 0.0171 | 0.048 | 0.354 | 0.724 | -0.078 | 0.112 |

| Omnibus: | 655.809 | Durbin-Watson: | 1.972 |
|---|---|---|---|
| Prob(Omnibus): | 0.000 | Jarque-Bera (JB): | 15924.397 |
| Skew: | -2.622 | Prob(JB): | 0.00 |
| Kurtosis: | 21.884 | Cond. No. | 167. |

Notes:
[1] Standard Errors assume that the covariance matrix of the errors is correctly specified.
  gl = gene length

**c**   Linear model for ksyn

| Dep. Variable: | ksyn | R-squared: | 0.077 |
|---|---|---|---|
| Model: | OLS | Adj. R-squared: | 0.071 |
| Method: | Least Squares | F-statistic: | 13.64 |
| Date: | Wed, 13 Mar 2024 | Prob (F-statistic): | 6.35e-15 |
| Time: | 15:25:01 | Log-Likelihood: | -447.86 |
| No. Observations: | 995 | AIC: | 909.7 |
| Df Residuals: | 988 | BIC: | 944.0 |
| Df Model: | 6 | | |

| | coef | std err | t | P > \|t\| | [0.025 | 0.975] |
|---|---|---|---|---|---|---|
| Intercept | 1.6326 | 0.126 | 12.937 | 0.000 | 1.385 | 1.880 |
| gl | -0.0526 | 0.029 | -1.794 | 0.073 | -0.110 | 0.005 |
| EPD_INR | -0.4811 | 0.231 | -2.086 | 0.037 | -0.934 | -0.028 |
| gl:EPD_INR | 0.1246 | 0.053 | 2.358 | 0.019 | 0.021 | 0.228 |
| EPD_TATA | 1.4663 | 0.415 | 3.535 | 0.000 | 0.652 | 2.280 |
| gl:EPD_TATA | -0.2797 | 0.104 | -2.696 | 0.007 | -0.483 | -0.076 |
| EPD_TATA:EPD_INR | 0.1158 | 0.123 | 0.941 | 0.347 | -0.126 | 0.357 |

| Omnibus: | 31.562 | Durbin-Watson: | 2.016 |
|---|---|---|---|
| Prob(Omnibus): | 0.000 | Jarque-Bera (JB): | 34.082 |
| Skew: | 0.453 | Prob(JB): | 3.97e-08 |
| Kurtosis: | 2.974 | Cond. No. | 167. |

Notes:
[1] Standard Errors assume that the covariance matrix of the errors is correctly specified.
  gl = gene length

**d**   Linear model for koff

| Dep. Variable: | koff | R-squared: | 0.014 |
|---|---|---|---|
| Model: | OLS | Adj. R-squared: | 0.008 |
| Method: | Least Squares | F-statistic: | 2.368 |
| Date: | Wed, 13 Mar 2024 | Prob (F-statistic): | 0.0281 |
| Time: | 15:21:58 | Log-Likelihood: | -112.42 |
| No. Observations: | 995 | AIC: | 238.8 |
| Df Residuals: | 988 | BIC: | 273.2 |
| Df Model: | 6 | | |

| | coef | std err | t | P > \|t\| | [0.025 | 0.975] |
|---|---|---|---|---|---|---|
| Intercept | 1.6743 | 0.090 | 18.588 | 0.000 | 1.498 | 1.851 |
| gl | 0.0345 | 0.021 | 1.648 | 0.100 | -0.007 | 0.075 |
| EPD_INR | -0.1902 | 0.165 | -1.155 | 0.248 | -0.513 | 0.133 |
| gl:EPD_INR | 0.0457 | 0.038 | 1.211 | 0.226 | -0.028 | 0.120 |
| EPD_TATA | -0.1520 | 0.296 | -0.513 | 0.608 | -0.733 | 0.429 |
| gl:EPD_TATA | 0.0415 | 0.074 | 0.560 | 0.575 | -0.104 | 0.187 |
| EPD_TATA:EPD_INR | 0.1414 | 0.088 | 1.610 | 0.108 | -0.031 | 0.314 |

| Omnibus: | 48.704 | Durbin-Watson: | 2.026 |
|---|---|---|---|
| Prob(Omnibus): | 0.000 | Jarque-Bera (JB): | 133.894 |
| Skew: | -0.191 | Prob(JB): | 8.42e-30 |
| Kurtosis: | 4.756 | Cond. No. | 167. |

Notes:
[1] Standard Errors assume that the covariance matrix of the errors is correctly specified.
  gl = gene length

**Extended Data Fig. 5 | Linear regression of core promoter elements and kinetic parameters.** Four independent linear regressions were made, to assess significant interactions between core promoter elements and each of these four quantities (**a**: burst size, **b**: burst frequency, **c**: *ksyn*, **d**: *koff*). The results of the modeling is presented, with pink rectangles highlighting the motifs, their interactions and p-values. The kinetic parameters was inferred from fibroblast new RNA, genes with controlled (that is low CI) burst size, kon, ksyn and koff. The results demonstrate that TATA motifs have significant effect on burst size and *ksyn*, but not with burst frequency and *koff*.

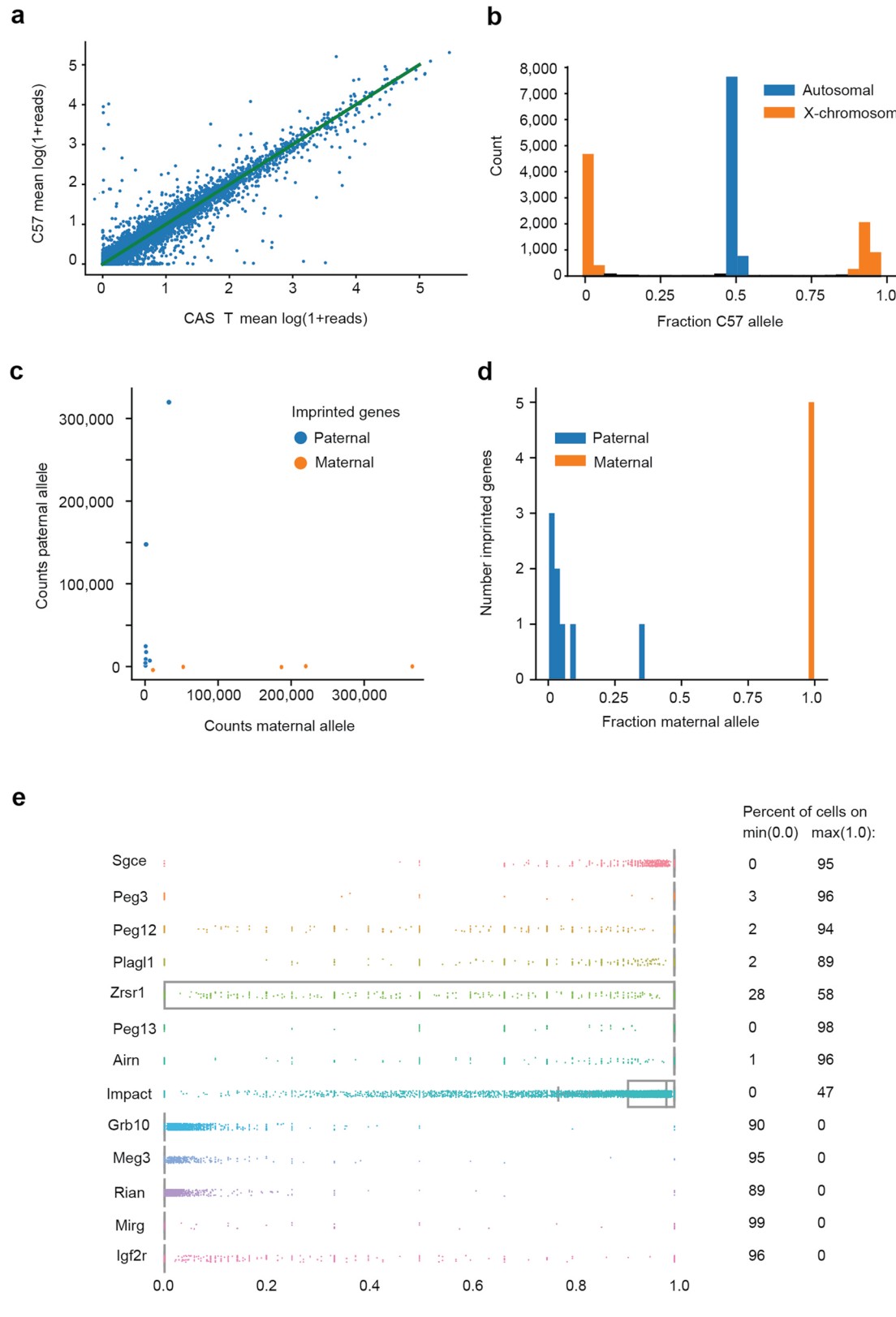

**Extended Data Fig. 6 | NASC-seq2 data accurately captures allelic regulation of X-chromosomal and imprinted genes. (a)** Scatter plots of total new RNA counts from the CAST and C57 alleles (mean across cells) showing the lack of overall bias in allelic counts. **(b)** Fraction of new RNA reads from the C57 allele for autosomal genes (blue) and genes on the X-chromosome (yellow), showing that overall new RNA counts for autosomal genes center on 0.5 and X-chromosome genes follow the inactivation. **(c)** Scatter plots showing with imprinted genes,

showing their total new RNA counts towards the maternal and paternal allele. **(d)** Histogram showing the fraction of reads assigned to the maternal allele for imprinted genes in primary fibroblast cells. **(e)** Fraction of reads per cells that were assigned to maternal origin for imprinted genes. The columns on the right show the percentage of cells that had 0 or 1 fraction of reads maternal. Source numerical data are available in source data.

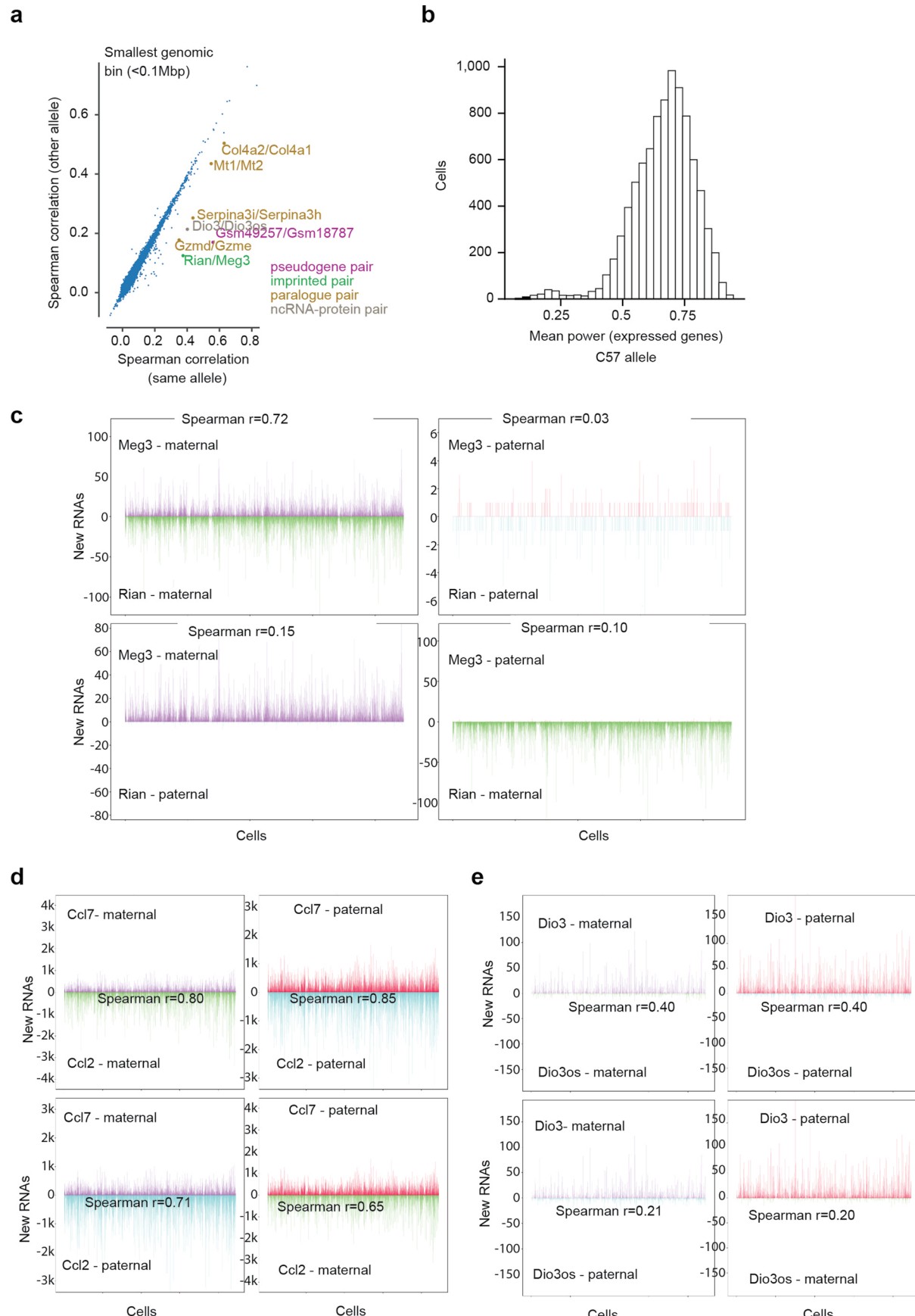

**a**

Smallest genomic bin (<0.1Mbp)

Col4a2/Col4a1
Mt1/Mt2
Serpina3i/Serpina3h
Dio3/Dio3os
Gsm49257/Gsm18787
Gzmd/Gzme
Rian/Meg3

pseudogene pair
imprinted pair
paralogue pair
ncRNA-protein pair

**Extended Data Fig. 7 | See next page for caption.**

**Extended Data Fig. 7 | Analysis of outlier gene-pairs with signs of co-bursting.**
(**a**) Scatter plot showing the spearman correlation for gene pairs based on new RNA counts from the same allele (x-axis) against the spearman correlation for gene pairs based on new RNA counts from the opposite alleles (y-axis), highlighting the strongest gene-pairs with increased spearman correlation only when based on new RNA counts from the same allele. Data shown is from the smallest genomic distance bin (< 0.1 Mbp). (**b**) Histogram showing the power to assign individual reads as new, for all reads assigned to the C57 allele. (**c**) Barplots showing the new RNA counts for the Meg3-Rian gene pair interaction (each shown on positive or negative y-axis scale), together with the observed Spearman correlations. For Meg3 and Rian, the increased correlation of new RNA counts from the same allele is simply a consequence of both genes having imprinting and only expression from the maternal allele, driving a non-meaningful signal of co-bursting. (**d**) Barplots showing the new RNA counts for the Ccl7-Ccl2 gene pair interaction (each shown on positive or negative y-axis scale), together with the observed Spearman correlations. For Ccl7 and Ccl2, the increased correlation of new RNA counts from the same allele could indicate a light increased co-bursting, although a technical nature such as read mismapping has not been ruled out. (**e**) Barplots showing the new RNA counts for the Dio3-Dio3os gene pair interaction (each shown on positive or negative y-axis scale), together with the observed Spearman correlations. For Dio3 and Dio3os, only Dio3 have high expression across the cells with very sparse counts for Dio3os – a non-coding RNA on the opposite strand of Dio3. Still the correlation in expression from the same allele is higher than the correlations from the opposite alleles, although in-depth scrutiny of read assignments to strands of these two overlapping RNAs have not been performed. Source numerical data are available in source data.

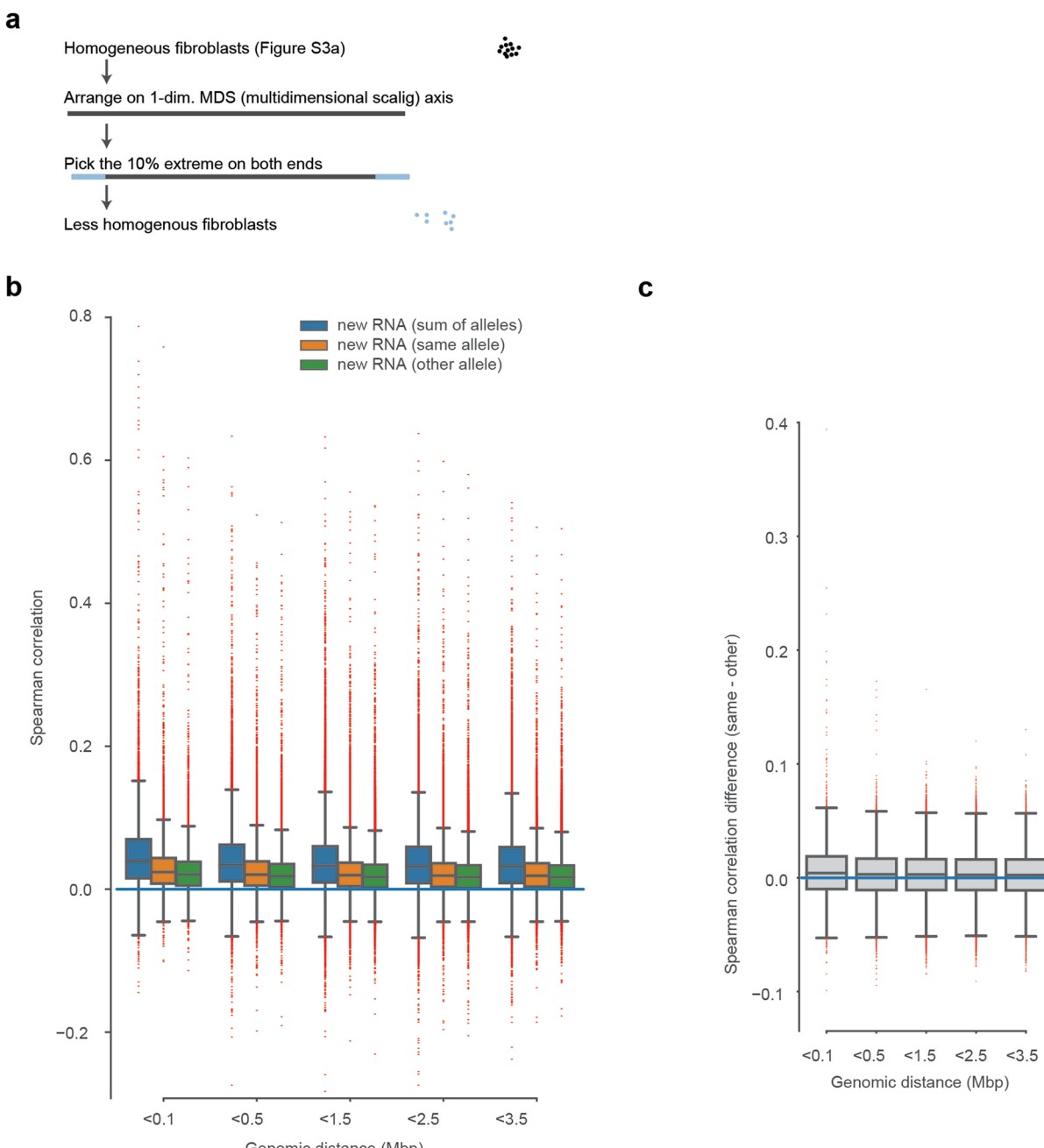

**Extended Data Fig. 8 | Effect of artificially introduced heterogeneity on co-bursting correlations.** (**a**) We turned our homogeneous fibroblasts data into a mix of two distinct groups by arranging the cells on a 1-dimensional similarity axis using multidimensional scaling and removing the middle 80% of cells. (**b**) The resulting data heterogeneity introduced a distance dependence of the gene-gene correlations, for both same allele, opposite allele and combined reads. (**c**) It also introduced a distance-dependent correlation difference between same allele and non-same (P = $10^{-10}$, Spearman correlation test). Significance of having a mean above zero for the first bin was P < $10^{-350}$ (Binomial test). Figures based on 208,379 gene-gene tests. Source numerical data are available in source data.

# Reporting Summary

## Statistics

For all statistical analyses, confirm that the following items are present in the figure legend, table legend, main text, or Methods section.

| n/a | Confirmed | |
|---|---|---|
| ☐ | ☒ | The exact sample size (*n*) for each experimental group/condition, given as a discrete number and unit of measurement |
| ☒ | ☐ | A statement on whether measurements were taken from distinct samples or whether the same sample was measured repeatedly |
| ☐ | ☒ | The statistical test(s) used AND whether they are one- or two-sided<br>*Only common tests should be described solely by name; describe more complex techniques in the Methods section.* |
| ☐ | ☒ | A description of all covariates tested |
| ☐ | ☒ | A description of any assumptions or corrections, such as tests of normality and adjustment for multiple comparisons |
| ☐ | ☒ | A full description of the statistical parameters including central tendency (e.g. means) or other basic estimates (e.g. regression coefficient) AND variation (e.g. standard deviation) or associated estimates of uncertainty (e.g. confidence intervals) |
| ☐ | ☒ | For null hypothesis testing, the test statistic (e.g. *F*, *t*, *r*) with confidence intervals, effect sizes, degrees of freedom and *P* value noted<br>*Give P values as exact values whenever suitable.* |
| ☒ | ☐ | For Bayesian analysis, information on the choice of priors and Markov chain Monte Carlo settings |
| ☒ | ☐ | For hierarchical and complex designs, identification of the appropriate level for tests and full reporting of outcomes |
| ☐ | ☒ | Estimates of effect sizes (e.g. Cohen's *d*, Pearson's *r*), indicating how they were calculated |

*Our web collection on statistics for biologists contains articles on many of the points above.*

## Software and code

Policy information about availability of computer code

| Data collection | No software was used for data collection. |
|---|---|
| Data analysis | Analysis of single-cell RNA-seq data was performed as described in detail in the methods. The procressing of Fastq files were carried out using zUMIs (v. 2.9.7), and aligned to the reference genome using STAR (v. 2.7.1 for human hg38 and v. 2.7.3a for mouse mm39 respectively). Analyses of 4sU base conversions in aligned sequenced data from NASC-seq2 were carried using custom Python code (available at github: https://github.com/sandberg-lab/NASC-seq2), as well as and further downstream analyses. |

For manuscripts utilizing custom algorithms or software that are central to the research but not yet described in published literature, software must be made available to editors and reviewers. We strongly encourage code deposition in a community repository (e.g. GitHub). See the Nature Portfolio guidelines for submitting code & software for further information.

## Data

Policy information about availability of data

All manuscripts must include a data availability statement. This statement should provide the following information, where applicable:
- Accession codes, unique identifiers, or web links for publicly available datasets
- A description of any restrictions on data availability
- For clinical datasets or third party data, please ensure that the statement adheres to our policy

Raw NASC-seq2 sequencing data (K562 and primary fibroblast cells) have been deposited in ENA (accession id: PRJEB60799) and source data has been deposited in

# Research involving human participants, their data, or biological material

Policy information about studies with human participants or human data. See also policy information about sex, gender (identity/presentation), and sexual orientation and race, ethnicity and racism.

| | |
|---|---|
| Reporting on sex and gender | Not applicable |
| Reporting on race, ethnicity, or other socially relevant groupings | Not applicable |
| Population characteristics | Not applicable |
| Recruitment | Not applicable |
| Ethics oversight | Not applicable |

Note that full information on the approval of the study protocol must also be provided in the manuscript.

# Field-specific reporting

Please select the one below that is the best fit for your research. If you are not sure, read the appropriate sections before making your selection.

☒ Life sciences          ☐ Behavioural & social sciences          ☐ Ecological, evolutionary & environmental sciences

For a reference copy of the document with all sections, see nature.com/documents/nr-reporting-summary-flat.pdf

# Life sciences study design

All studies must disclose on these points even when the disclosure is negative.

| | |
|---|---|
| Sample size | Sample sizes were not predetermined using statistical analysis, however the power to infer new and old transcripts based on length of RNA sequencing was determined as shown in the study. |
| Data exclusions | Single-cell RNA-seq data were filtered according to established criteria. Cutoffs are listed where appropriate. |
| Replication | All experiments were performed across hundreds to tens of thousands of individual cells, and kinetic analyses from subsets of cells showed in general good agreement. |
| Randomization | Cells were randomly sorted into microwell plates using FACS, 4sU and no-4sU control cells, present on each plate when possible. |
| Blinding | Investigators were not blinded to groups of samples, since we only analyzed one type of mouse fibroblasts and one group of human K562 cells. |

# Reporting for specific materials, systems and methods

We require information from authors about some types of materials, experimental systems and methods used in many studies. Here, indicate whether each material, system or method listed is relevant to your study. If you are not sure if a list item applies to your research, read the appropriate section before selecting a response.

## Materials & experimental systems

| n/a | Involved in the study |
|---|---|
| ☒ | ☐ Antibodies |
| ☐ | ☒ Eukaryotic cell lines |
| ☒ | ☐ Palaeontology and archaeology |
| ☐ | ☒ Animals and other organisms |
| ☒ | ☐ Clinical data |
| ☒ | ☐ Dual use research of concern |
| ☒ | ☐ Plants |

## Methods

| n/a | Involved in the study |
|---|---|
| ☒ | ☐ ChIP-seq |
| ☒ | ☐ Flow cytometry |
| ☒ | ☐ MRI-based neuroimaging |

# Eukaryotic cell lines

Policy information about cell lines and Sex and Gender in Research

| | |
|---|---|
| Cell line source(s) | K562 cells from DSMZ (ACC-10). |
| Authentication | K562 cells were authenticated at the DSMZ Identification Service according to standards for STR profiling (ASN-0002). |
| Mycoplasma contamination | Cells were tested for Mycoplasma and were confirmed negative (MycoAlert, Lonza). |
| Commonly misidentified lines (See ICLAC register) | No commonly misidentified lines were used in this study. |

# Animals and other research organisms

Policy information about studies involving animals; ARRIVE guidelines recommended for reporting animal research, and Sex and Gender in Research

| | |
|---|---|
| Laboratory animals | Primary mouse fibroblasts were obtained from the tail of a 5 month old female CAST/EiJ x C57BL/6J mouse, as described in the metods section. Animals were housed in standard housing conditions (ambient temperature of 20-22°C and humidity of 40-60%) with 12:12-hour light:dark cycles with food and water ad libitum. All experimental procedures were approved by the Stockholms Norra Djurförsöksetiska Nämnd. |
| Wild animals | No wild animals were used in this study. |
| Reporting on sex | Large experiment on primary mouse fibroblasts were performed on a female (in order to have X-chromosome inactivation as an allellic control), the conclusions on general transcriptional bursting and co-bursting should not differ substantially between sexes. |
| Field-collected samples | No field-collected samples were used in this study. |
| Ethics oversight | Ethical permit numbers N95/15 and 13572-2020 from Jordbruksverket (Sweden). |

Note that full information on the approval of the study protocol must also be provided in the manuscript.

