## [Peer Review File · Nature Cell Biology]

Peer Review Information

Journal: Nature Cell Biology

Manuscript Title: Single-cell new RNA sequencing reveals principles of transcription at the resolution of individual bursts

Corresponding author name(s): Professor Rickard Sandberg

Editorial Notes:

Transferred manuscripts This manuscript has been previously reviewed at another journal that is not operating a transparent peer review scheme. This document only contains reviewer comments, rebuttal and decision letters for versions considered at Nature Cell Biology.

Reviewer Comments & Decisions:

Decision Letter, initial version:

Our ref: NCB-LE53741-T

25th April 2024

Dear Dr. Sandberg,

Thank you for submitting your revised manuscript "Single-cell new RNA sequencing reveals principles of transcription at the resolution of individual bursts" (NCB-LE53741-T), with sincere apologies for the delay in sending a decision. It has now been seen by the original referees and their comments are below. The reviewers find that the paper has improved in revision, and therefore we'll be happy in principle to publish it in Nature Cell Biology, pending minor revisions to satisfy the referees' final requests and to comply with our editorial and formatting guidelines.

**Please note that we no longer publish the letter format, and we would consider this manuscript in our article format.

**Our articles have 5-8 main figures and all key data should be presented in these figures. Articles can have up to 10 extended data figures. All key data should be presented in the main figures, with Extended Data only presenting supportive information. Please convert all Supplementary Figures into

Extended Data Figures. Please keep the current structure of the figures, but ensure that you are using the full A4 space and increase panel size and font size (at least 7 pt) for improved readability. More data can be moved from Extended Data Figures into the main figures. No peer-reviewed data should be removed.

We are now performing detailed checks on your paper and will send you a checklist detailing our editorial and formatting requirements in about a week.

**Please do not upload the final materials and make any revisions until you receive this additional information from us.

Thank you again for your interest in Nature Cell Biology Please do not hesitate to contact me if you have any questions.

Sincerely,

Sabrya Carim, PhD
(she/her/hers)
Associate Editor, Nature Cell Biology
Nature Portfolio

Springer Nature
The Campus, 4 Crinan Street, London N1 9XW, UK
sabrya.carim@springernature.com
<https://orcid.org/0000-0001-9485-1938>

Reviewer #1 (Remarks to the Author):

I had already indicated in my previous report that I consider the manuscript now as suitable for publication, but I had made a few suggestion for minor improvements, which the authors followed by making a number of suitable wording changes and adding one analysis. Therefore, I fully recommend to publish the paper now as is.

Note: Following Nature Publishing's new scheme of optionally de-anonymizing reviewer comments for enhanced transparency, I declare that I consent with the publication of this and my preceding reviewer reports, and disclose my identity.

-- Simon Anders, University of Heidelberg

Decision Letter, final checks:

Our ref: NCB-LE53741-T

10th May 2024

Dear Dr. Sandberg,

Thank you for your patience as we've prepared the guidelines for final submission of your Nature Cell Biology manuscript, "Single-cell new RNA sequencing reveals principles of transcription at the resolution of individual bursts" (NCB-LE53741-T). Please carefully follow the step-by-step instructions provided in the attached file, and add a response in each row of the table to indicate the changes that you have made. Ensuring that each point is addressed will help to ensure that your revised manuscript can be swiftly handed over to our production team.

In recognition of the time and expertise our reviewers provide to Nature Cell Biology's editorial process, we would like to formally acknowledge their contribution to the external peer review of your manuscript entitled "Single-cell new RNA sequencing reveals principles of transcription at the resolution of individual bursts". For those reviewers who give their assent, we will be publishing their names alongside the published article.

Nature Cell Biology offers a Transparent Peer Review option for new original research manuscripts submitted after December 1st, 2019. As part of this initiative, we encourage our authors to support increased transparency into the peer review process by agreeing to have the reviewer comments, author rebuttal letters, and editorial decision letters published as a Supplementary item. When you submit your final files please clearly state in your cover letter whether or not you would like to participate in this initiative. Please note that failure to state your preference will result in delays in accepting your manuscript for publication.

Cover suggestions

COVER ARTWORK: We welcome submissions of artwork for consideration for our cover. For more information, please see our guide for cover artwork.

Nature Cell Biology has now transitioned to a unified Rights Collection system which will allow our Author Services team to quickly and easily collect the rights and permissions required to publish your work. Approximately 10 days after your paper is formally accepted, you will receive an email in providing you with a link to complete the grant of rights. If your paper is eligible for Open Access, our

Author Services team will also be in touch regarding any additional information that may be required to arrange payment for your article.

Please note that *Nature Cell Biology* is a Transformative Journal (TJ). Authors may publish their research with us through the traditional subscription access route or make their paper immediately open access through payment of an article-processing charge (APC). Authors will not be required to make a final decision about access to their article until it has been accepted. Find out more about Transformative Journals

[Redacted]

Best regards,

Kendra Donahue
Staff
Nature Cell Biology

On behalf of

Sabrya Carim, PhD
(she/her/hers)
Associate Editor, Nature Cell Biology
Nature Portfolio

Springer Nature
The Campus, 4 Crinan Street, London N1 9XW, UK
sabrya.carim@springernature.com

<https://orcid.org/0000-0001-9485-1938>

Reviewer #1:

Remarks to the Author:

I had already indicated in my previous report that I consider the manuscript now as suitable for publication, but I had made a few suggestion for minor improvements, which the authors followed by making a number of suitable wording changes and adding one analysis. Therefore, I fully recommend to publish the paper now as is.

Note: Following Nature Publishing's new scheme of optionally de-anonymizing reviewer comments for enhanced transparency, I declare that I consent with the publication of this and my preceding reviewer reports, and disclose my identity.

-- Simon Anders, University of Heidelberg

Final Decision Letter:

Dear Dr Sandberg,

I am pleased to inform you that your manuscript, "Single-cell new RNA sequencing reveals principles of transcription at the resolution of individual bursts", has now been accepted for publication in Nature Cell Biology. Congratulations!

Please note that *Nature Cell Biology* is a Transformative Journal (TJ). Authors may publish their research with us through the traditional subscription access route or make their paper immediately open access through payment of an article-processing charge (APC). Authors will not be required to make a final decision about access to their article until it has been accepted. Find out more about Transformative Journals

If you have not already done so, we strongly recommend that you upload the step-by-step protocols used in this manuscript to protocols.io (<https://protocols.io>), an open online resource that allows researchers to share their detailed experimental know-how. All uploaded protocols are made freely

available and are assigned DOIs for ease of citation. Protocols and Nature Portfolio journal papers in which they are used can be linked to one another, and this link is clearly and prominently visible in the online versions of both. Authors who performed the specific experiments can act as primary authors for the Protocol as they will be best placed to share the methodology details, but the Corresponding Author of the present research paper should be included as one of the authors. By uploading your Protocols onto protocols.io, you are enabling researchers to more readily reproduce or adapt the methodology you use, as well as increasing the visibility of your protocols and papers. You can also establish a dedicated workspace to collect your lab Protocols. Further information can be found at <https://www.protocols.io/help/publish-articles>.

With kind regards,

Sabrya Carim, PhD
(she/her/hers)
Associate Editor, Nature Cell Biology
Nature Portfolio

Springer Nature
The Campus, 4 Crinan Street, London N1 9XW, UK
sabrya.carim@springernature.com
<https://orcid.org/0000-0001-9485-1938>

** Visit the Springer Nature Editorial and Publishing website at www.springernature.com/editorial-and-publishing-jobs for more information about our career opportunities. If you have any questions please click here.**